# Bridgin connects the outer kinetochore to centromeric chromatin

Shreyas Sridhar [1,4], Tetsuya Hori[2], Reiko Nakagawa [3], Tatsuo Fukagawa [2✉] & Kaustuv Sanyal [1,2✉]

The microtubule-binding outer kinetochore is coupled to centromeric chromatin through CENP-C[Mif2], CENP-T[Cnn1], and CENP-U[Ame1] linker pathways originating from the constitutive centromere associated network (CCAN) of the inner kinetochore. Here, we demonstrate the recurrent loss of most CCAN components, including certain kinetochore linkers during the evolution of the fungal phylum of Basidiomycota. By kinetochore interactome analyses in a model basidiomycete and human pathogen *Cryptococcus neoformans*, a forkhead-associated domain containing protein "bridgin" was identified as a kinetochore component along with other predicted kinetochore proteins. In vivo and in vitro functional analyses of bridgin reveal its ability to connect the outer kinetochore with centromeric chromatin to ensure accurate chromosome segregation. Unlike established CCAN-based linkers, bridgin is recruited at the outer kinetochore establishing its role as a distinct family of kinetochore proteins. Presence of bridgin homologs in non-fungal lineages suggests an ancient divergent strategy exists to bridge the outer kinetochore with centromeric chromatin.

[1] Molecular Mycology Laboratory, Molecular Biology and Genetics Unit, Jawaharlal Nehru Center for Advanced Scientific Research (JNCASR), Bangalore, India 560064. [2] Laboratory of Chromosome Biology, Graduate School of Frontier Biosciences, Osaka University, Suita, Osaka 565-0871, Japan. [3] Laboratory for Phyloinformatics, RIKEN Center for Biosystems Dynamics Research (BDR), Kobe, Japan. [4] Present address: Graduate School of Frontier Biosciences, Osaka University, Suita, Osaka 565-0871, Japan. ✉email: tfukagawa@fbs.osaka-u.ac.jp; sanyal@jncasr.ac.in

Accurate chromosome segregation ensures faithful transmission of the genetic material to progeny. The kinetochore is a multicomplex protein structure assembled on centromere DNA of each eukaryotic chromosome[1–4] and is attached to the spindle microtubules for accurate chromosome segregation[5]. Components involved in error correction mechanisms, including the spindle assembly checkpoint (SAC), are recruited at kinetochores to ensure the biorientation of sister chromatids in mitosis[6–8]. The inner kinetochore is composed of centromeric histone H3 variant CENP-A[Cse4] [9–11] and the 16-member constitutive centromere-associated network (CCAN)[12–16]. The outer kinetochore members of the KMN (Knl1, Mis12, and Ndc80 complexes) network[1,17] are recruited to CCAN to form the kinetochore ensemble across eukaryotes, including budding yeast and vertebrates[18]. Additional components such as the 10-member Dam1 complex (Dam1C) in fungi and the three-member Ska complex in vertebrates localize to the outer kinetochore to ensure accurate kinetochore–microtubule interactions[19–24].

Genome sequencing data reveal that while the KMN network components are highly conserved[25–27], the inner kinetochore composition is variable across eukaryotes[25–29]. The Ndc80 complex (Ndc80C) of the KMN network directly binds to spindle microtubules[30–32]. CCAN proteins, CENP-C[Mif2] and CENP-T[Cnn1], have been shown to independently link centromeric chromatin with the KMN network[33–38]. In addition, CENP-U[Ame1] functions as a linker in budding yeast[39,40]. CENP-C[Mif2] and CENP-T[Cnn1], through their amino(N)-termini, interact with the KMN network, while their carboxy(C)-termini interact with centromeric chromatin[41–43]. CENP-U[Ame1] cooperatively with CENP-C[Mif2] also binds to Mis12C, to ensure a linker function in budding yeast[39,40]. These three critical kinetochore linker proteins, CENP-C[Mif2], CENP-T[Cnn1], and CENP-U[Ame1], are often lost or significantly diverged during evolution. Despite the observed loss or functional divergence of linker proteins, the existence of other molecular innovations to bridge centromeric chromatin with the outer kinetochore to ensure accurate chromosome segregation remains largely unknown[21,28,44]. Although each of the kinetochore linker pathways has been shown to play important roles in chromosome segregation, the architectural dependence on a specific pathway varies across species[34,35,37].

*Cryptococcus neoformans* is a ubiquitous environmental fungus and an opportunistic pathogen causing fatal cryptococcal meningitis[45,46]. Our previous studies using this model basidiomycete suggested a stepwise kinetochore assembly on a long repetitive regional centromere[24,47].

In this work, by analyzing a number of fungal genomes, we strikingly find the recurrent absence of most CCAN components, including certain conventional linker proteins, CENP-T[Cnn1] and CENP-U[Ame1], in the phylum Basidiomycota. We identify "bridgin", a previously undescribed protein, in the kinetochore interactome of *C. neoformans*. We predict the existence of bridgin homologs in eukaryotic lineages outside the fungal kingdom as well. Based on in vivo and in vitro functional analyses, we demonstrate that bridgin connects the outer kinetochore with centromeric chromatin revealing the existence of a distinct strategy to bridge the outer kinetochore and centromeric chromatin across eukaryotes.

## Results

**Recurrent independent loss events of CCAN proteins in Basidiomycota.** To have a comprehensive understanding of the kinetochore composition in Basidiomycota, we analyzed putative kinetochore homologs using high-confidence protein homology searches combined with secondary and tertiary structure prediction. Species representing 31 fungal orders across the three subphyla, Pucciniomycotina, Ustilagomycotina, and Agaricomycotina, were considered. Additionally, representative species across 7 fungal phyla were included[48]. CENP-A[Cse4], the 16-member CCAN, and the 10-member KMN network were chosen for this study (Fig. 1a and Supplementary Data 1). Our analysis, in accordance with the previous reports[25,26,29], indicates the robust conservation of CENP-A[Cse4] and the KMN network proteins across basidiomycetes. On the other hand, we observed recurrent loss of most CCAN proteins across 23 of the 31 basidiomycete orders (Fig. 1a). In the subphylum of Agaricomycotina, to which *C. neoformans* belongs, the loss event may have occurred early at the time of divergence of Wallemiales from other orders. Retention of CCAN proteins in a few discrete orders in the subphyla of Pucciniomycotina and Ustilogamycotina suggests the occurrence of multiple independent loss events of most CCAN subunits (Fig. 1a). The kinetochore linker CENP-C[Mif2] was the only conserved CCAN component across basidiomycetes. Other known linker proteins, CENP-T[Cnn1] and CENP-U[Ame1], were often lost together. Although the primary protein sequence conservation is low among CENP-T[Cnn1] homologs, they share a typical protein architecture: an N-terminal α-helix composed of conserved hydrophobic residues, observed to occur within the PITH domain in Ustilaginales, and the CENP-T[Cnn1] motif at the C terminus (Fig. 1b). As a result, the CENP-C[Mif2] pathway remains the only known linker pathway among 23 of the 31 basidiomycete orders investigated. These observations suggest that additional factors may exist to compensate for the recurrent loss of most CCAN proteins in Basidiomycota.

**Identification of the kinetochore interactome in *C. neoformans*.** In search of such unknown kinetochore proteins, we attempted to analyze the kinetochore composition in the fungal phylum of Basidiomycota using a relatively well-studied model basidiomycetous yeast *C. neoformans* (Fig. 1a). To comprehensively determine the constitution of the kinetochore and its interactome in vivo, *C. neoformans* strains expressing 3×FLAG-tagged CENP-C[Mif2], Dsn1 (Mis12C), and Spc25 (Ndc80C) under control of the native promoter were generated. The functionality of the 3×FLAG-tagged proteins was validated by chromatin immunoprecipitation (ChIP) and co-immunoprecipitation (co-IP) assays (Supplementary Fig. 1a–c).

Mass-spectrometry (MS) analyses were performed after FLAG-IP for CENP-C[Mif2], Dsn1, and Spc25 from metaphase-enriched (mitotic index > 90%) cell population (Fig. 1c and Supplementary Data 2). Nearly all predicted inner kinetochore proteins (CENP-A[Cse4] and CENP-C[Mif2]) and outer kinetochore KMN network components were identified from each of the FLAG-IPs (Table 1 and Supplementary Data 2). Except for CENP-C[Mif2], no components of the CCAN were identified in the IP–MS (Table 1 and Supplementary Table. 2). Identified KMN network components included evolutionarily conserved but previously unannotated ORFs coding for subunits of Mis12C (CNAG_04300[Nsl1], CNAG_04479[Nnf1]) and Knl1C (CNAG_03715[Sos7]) (Table 1 and Supplementary Data 2). CNAG_03715[Sos7] (Knl1C) was used as a candidate to authenticate the predicted ORFs as bona fide kinetochore proteins. CNAG_03715[Sos7] was bioinformatically identified as the Sos7 homolog in *C. neoformans* using *Schizosaccharomyces pombe* Sos7[49] (Supplementary Fig. 1d, e). Microscopic observations of GFP-CnSos7 suggested a cell-cycle-dependent kinetochore-exclusive localization in *C. neoformans* (Fig. 1d and Supplementary Fig. 1f). Further analysis confirmed that while Sos7 was dispensable for cell viability, yet was essential for accurate chromosome segregation in *C. neoformans* (Supplementary Fig. 1g, h).

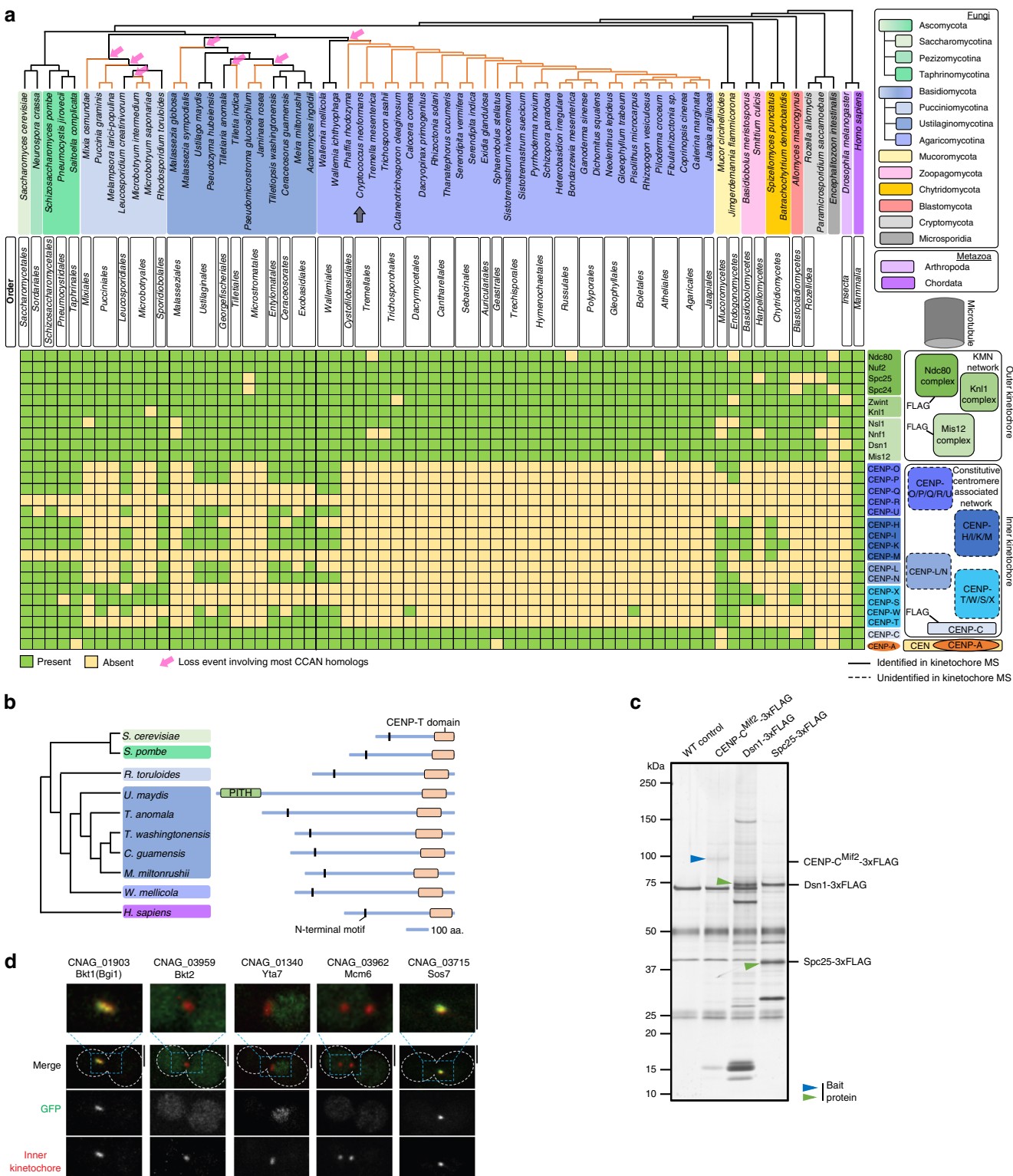

Taken together, the experimental identification of the kinetochore interactome confirms our bioinformatics prediction that *C. neoformans* retains a single known kinetochore linker pathway through CENP-C$^{Mif2}$.

**Bridgin (Bgi1) is a basidiomycete kinetochore protein**. Next, we attempted to find the existence of unknown kinetochore proteins compensating for the loss of CCAN subunits in Basidiomycota.

We gained confidence by identifying nearly all known structural kinetochore components from each FLAG-IP–MS experiment. Thus, we hypothesized that it was possible to identify previously undescribed kinetochore proteins, if any, from the identified common interactors of CENP-C$^{Mif2}$, Dsn1, and Spc25 (Supplementary Data 2). Among the identified proteins that fulfill this stringent criterion, we categorized two sets of proteins: (a) conserved among basidiomycetes with unknown function and

**Fig. 1 Identification of the kinetochore interactome in *Cryptococcus neoformans*. a** Conservation of kinetochore proteins across the mentioned species. The cladogram represents the relationship between the species; each subphylum is color-coded. The presence (green boxes) or absence (yellow boxes) of each kinetochore protein in every species is shown. Pink arrows indicate loss events of CCAN proteins, excluding CENP-C[Mif2]. Orange lines in the cladogram refer to basidiomycete lineages that have lost most CCAN components. The gray arrow points to *C. neoformans. Right*, schematic of a typical kinetochore ensemble. FLAG-labeled proteins of kinetochore subcomplexes (CENP-C, Mis12C, and Ndc80C) in *C. neoformans* are indicted and used for the immunoprecipitation (IP)–mass-spectrometry (MS) identification of the kinetochore interactome. Protein complexes identified (solid borders) or those that remained unidentified (dotted borders) by the IP–MS experiment in *C. neoformans* are shown. **b** Domain architectures of identified CENP-T[Cnn1] homologs among basidiomycetes. Color coding of subphyla is followed as in **a. c** Proteins eluted after FLAG-IP, of CENP-C[Mif2] in the *C. neoformans* strain SHR896 (*CENP-C::CENP-C[MIF2]-3×FLAG*), Dsn1 in SHR824 (*DSN1::DSN1-3×FLAG*), Spc25 in SHR823 (*SPC25::SPC25-3×FLAG*), and an untagged wild-type (WT) control (H99) after thiabendazole (TBZ) treatment, were separated on a gradient PAGE gel and silver-stained. Blue and green arrows mark the inner and outer kinetochore bait proteins, respectively. **d** Micrographs of *C. neoformans* cells at the M phase expressing GFP-tagged proteins identified by the IP–MS screen. Kinetochores are marked by inner kinetochore proteins, CENP-C[Mif2]-mCherry for Bgi1[Bkt1] in SHR876 (*BGI1::BGI1-V5-GFP*), Bkt2 in SHR897 (*BKT2::BKT2-V5-GFP*), and Yta7 in SHR842 (*YTA7::YTA7-V5-GFP*) or mCherry-CENP-A[Cse4] for Mcm6 in SHR905 (*MCM6::MCM6-V5-GFP*) and Sos7 in SHR845 (*SOS7::GAL7p-GFP-SOS7*). Scale bar, 3 µm. Source data are available as a Source Data file.

**Table 1 List of kinetochore proteins obtained in the kinetochore IP–MS.**

| Complex | Protein | WT control | | CENP-C[Mif2]–3×FLAG | | Dsn1–3xFLAG | | Spc25-3×FLAG | |
|---|---|---|---|---|---|---|---|---|---|
| | | C (%) | TSC | C (%) | TSC | C (%) | TSC | C (%) | TSC |
| Ndc80 | Ndc80 | – | – | 16 | 8 | 46 | 113 | 61 | 1130 |
| | Nuf2 | 2.7 | 1 | 10 | 4 | 44 | 67 | 69 | 683 |
| | Spc25 | 19 | 7 | 21 | 6 | 64 | 90 | 84 | 581 |
| | Spc24 | – | – | 23 | 6 | 73 | 51 | 82 | 313 |
| Knl1 | Sos7 (CNAG_03715) | – | – | 11 | 4 | 55 | 74 | 57 | 32 |
| | Knl1[Spc105] | – | – | 8 | 9 | 31 | 272 | 38 | 206 |
| Mis12 | Nsl1 (CNAG_04300) | – | – | 15 | 10 | 67 | 84 | 47 | 24 |
| | Nnf1 (CNAG_04479) | – | – | 12 | 6 | 52 | 222 | 39 | 68 |
| | Dsn1 | – | – | 26 | 20 | 61 | 607 | 48 | 127 |
| | Mis12[Mtw1] | – | – | 30 | 11 | 72 | 287 | 49 | 56 |
| CCAN | CENP-C[Mif2] | – | – | 48 | 272 | 35 | 66 | – | – |
| CENP-A | CENP-A[Cse4] | – | – | 24 | 9 | 20 | 10 | 6 | 2 |

Percentage of amino acid sequence coverage (C,%) and the total spectrum counts (TSC) specific to the corresponding protein obtained by IP-MS are mentioned.

termed them basidiomycete kinetochore proteins (Bkts), which included CNAG_01903[Bkt1], CNAG_03959[Bkt2], and CNAG_02701[Bkt3] (Supplementary Fig. 1i), and (b) known chromatin-interacting proteins with uncharacterized kinetochore function, that included CNAG_01340[Yta7] and all components of the evolutionarily conserved DNA replication initiation factors Mcm2–7 of the MCM complex[50] (Supplementary Fig. 1j). We characterized CNAG_03962[Mcm6] as the representative test candidate for the MCM complex. As a secondary screen, each of these five candidates was expressed as C-terminally V5-GFP epitope-tagged proteins from their endogenous locus. We could observe subcellular localization in strains expressing CNAG_01903[Bkt1], CNAG_03959[Bkt2], CNAG_01340[Yta7], or CNAG_03962[Mcm6]. The protein encoded by CNAG_01903[Bkt1] colocalized at the kinetochore with CENP-C[Mif2] (Fig. 1d and Supplementary Fig. 1k). Other proteins did not show exclusive kinetochore localization, although some puncta of CNAG_01340[Yta7] and CNAG_03962[Mcm6] colocalized transiently with inner kinetochore proteins (CENP-C[Mif2] or CENP-A[Cse4]) at G1/S stage (Supplementary Fig. 1k). Recently, Yta7 was shown to act as a deposition factor for CENP-A[Cse4] [51]. Based on the localization dynamics of Bkts, CNAG_01903[Bkt1] was taken forward as a bona fide candidate kinetochore protein due to its exclusive kinetochore localization. Considering its identified function through this study, we refer to Bkt1 as "bridgin" (Bgi1) henceforth.

**Multiple outer kinetochore receptors recruit bridgin.** Cross-linked ChIP analysis suggested that bridgin localized to the centromere at M phase but absent at S phase (Fig. 2a and Supplementary Fig. 2a). We validated the localization by microscopic

observations (Supplementary Fig. 1k). Cellular pools of bridgin were relatively uniform through the cell cycle, suggesting that the G2/M kinetochore localization of bridgin is not due to a higher level of protein expression at the corresponding stage (Supplementary Fig. 2b). Further, bridgin localization at the kinetochore was not affected by the treatment of a microtubule-depolymerizing drug, thiabendazole (TBZ) (Supplementary Fig. 2c), suggesting that loss of spindle integrity did not influence the kinetochore localization of bridgin.

Our previous study suggested a stepwise assembly of the kinetochore in *C. neoformans*, yet no sequential assembly of kinetochore subcomplexes was established[24]. We investigated where bridgin localizes within the kinetochore localization hierarchy. We adopted a microscopy-based kinetochore interdependency assay strategy involving a test kinetochore protein in the background of a conditional kinetochore mutant[52] (Fig. 2b). Conditional mutants of CENP-A[Cse4], CENP-C[Mif2], Mis12[Mtw1], Nuf2, Knl1[Spc105], Dad1, and Dad2 in *C. neoformans* were generated by driving expression of each of these GFP/mCherry-tagged kinetochore proteins under the control of the *GAL7* promoter (*GAL7p*). Upon reduced protein levels due to nonpermissive conditions of the *GAL7p*-driven expression (Supplementary Fig. 2d, e), the kinetochore mutants displayed a lack of growth (Supplementary Fig. 2f) on account of reduced cell viability (Supplementary Fig. 2g). These results, in accordance with our previous report[53], validated the effective repression mediated by *GAL7p*. Using this experimental strategy, we determined that Mis12C and Ndc80C influence the stability of each other at the kinetochore (Supplementary Fig. 3a, b). Knl1C (Supplementary Fig. 3c–f) and Dam1C (Supplementary Fig. 3g–i)) independently require the Mis12C–Ndc80C platform to be localized

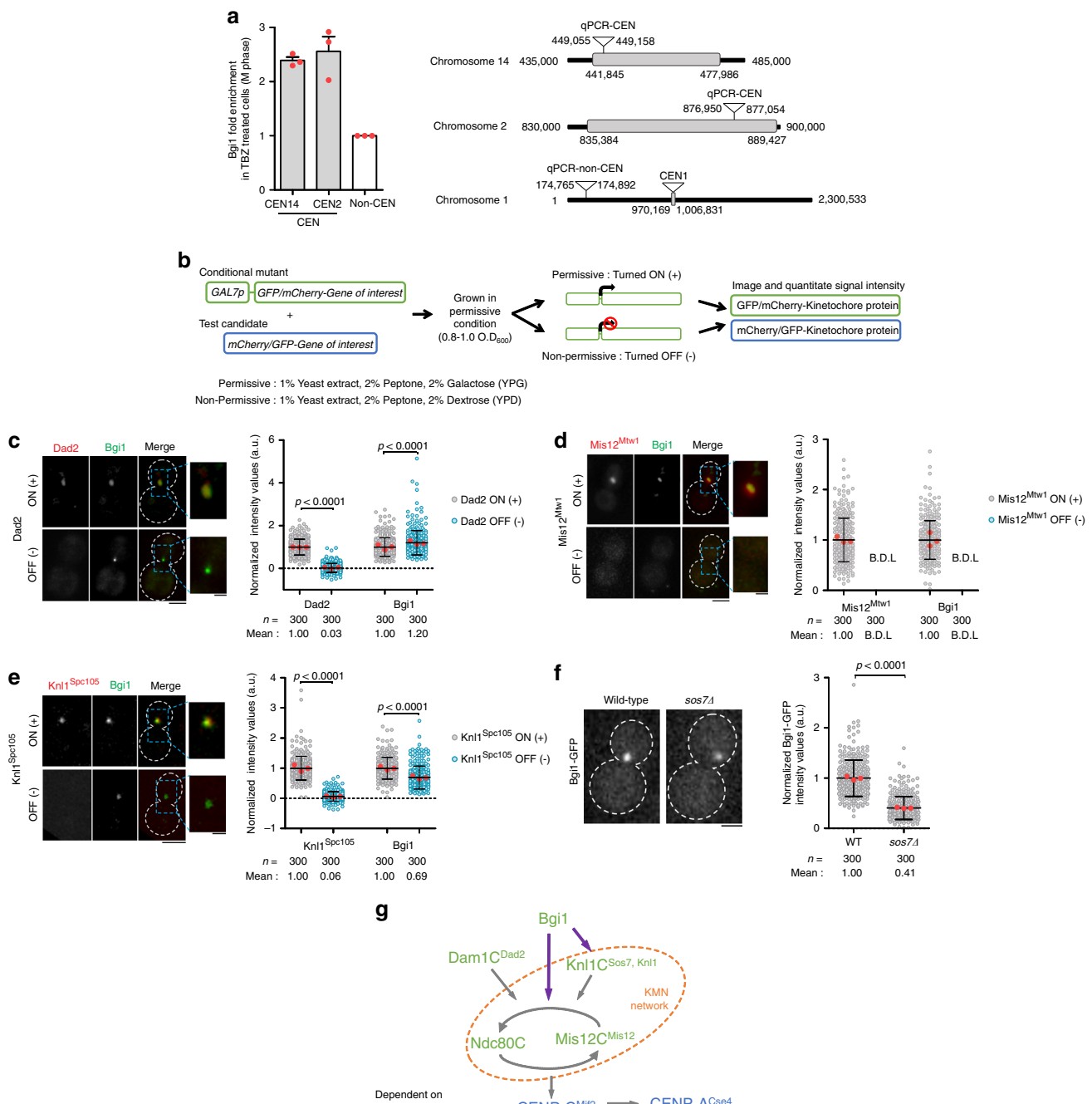

**Fig. 2 Bridgin assembles onto the KMN network platform. a** *Left*, enrichment of Bgi1 at M-phase kinetochores in SHR870 (*BGI1::BGI1-V5-GFP*) cells by cross-linked chromatin immunoprecipitation (ChIP) followed by quantitative polymerase chain reaction (qPCR). The data represent the mean ± standard deviation (S.D.) of three independent experiments. *Right*, schematic representing the location of qPCR primers used. Centromeric sequences are defined by grey boxes. **b** Steps followed to determine localization interdependency at the kinetochore among its subcomplexes. **c**–**e** *Left*, a representative image of Bgi1-V5-GFP in **c** SHR906 (*DAD2::GAL7p-mCherry-DAD2*), **d** SHR907 (*MIS12^MTW1^::GAL7p-mCherry-MIS12^MTW1^*) and **e** SHR945 (*KNL1^SPC105^::GAL7p-mCherry-KNL1^SPC105^*) in permissive (+) or nonpermissive (−) conditions of the *GAL7p*. Scale bar, 3 μm; 1 μm for magnified inset. *Right*, Bgi1-V5-GFP signals and **c** mCherry-Dad2, **d** mCherry-Mis12^Mtw1^, or **e** mCherry-Knl1^Spc105^ in cells with or without Dad2, Mis12^Mtw1^, or Knl1^Spc105^ expression, respectively, were quantified and normalized to their respective mean signals in the permissive condition of the *GAL7p*. Signals were measured after 18, 12, and 18 h in nonpermissive conditions for Dad2, Mis12^Mtw1^, and Knl1^Spc105^, respectively. For **d**, the strong influence of Mis12^Mtw1^ on kinetochore localization of bridgin resulted in signals that were below detectable levels (B.D.L.). For **e**, bridgin signals in 24 out of 300 cells were below detectable levels. **f** *Left*, representative images of Bgi1-V5-GFP in wild-type H99 (*SOS7*) or in *sos7* null mutant SHR908 (*SOS7::sos7Δ*) cells. Scale bar, 2 μm. *Right*, Bgi1-V5-GFP signal intensities were quantified and normalized to the mean wild-type signal. **c**–**f** The results of three independent experiments with 100 cells each are represented. The red dot represents the mean of one experiment; mean ± S.D. is shown. For statistical comparison of differences between the samples, Mann–Whitney two-tailed analysis was applied, *p* values show significant differences. **g** Schematic describes the observed localization interdependency of inner (blue) and outer (green) kinetochore protein complexes at the *C. neoformans* kinetochore. Purple arrows highlight the dependency of bridgin on the KMN network. Source data are available as a Source Data file.

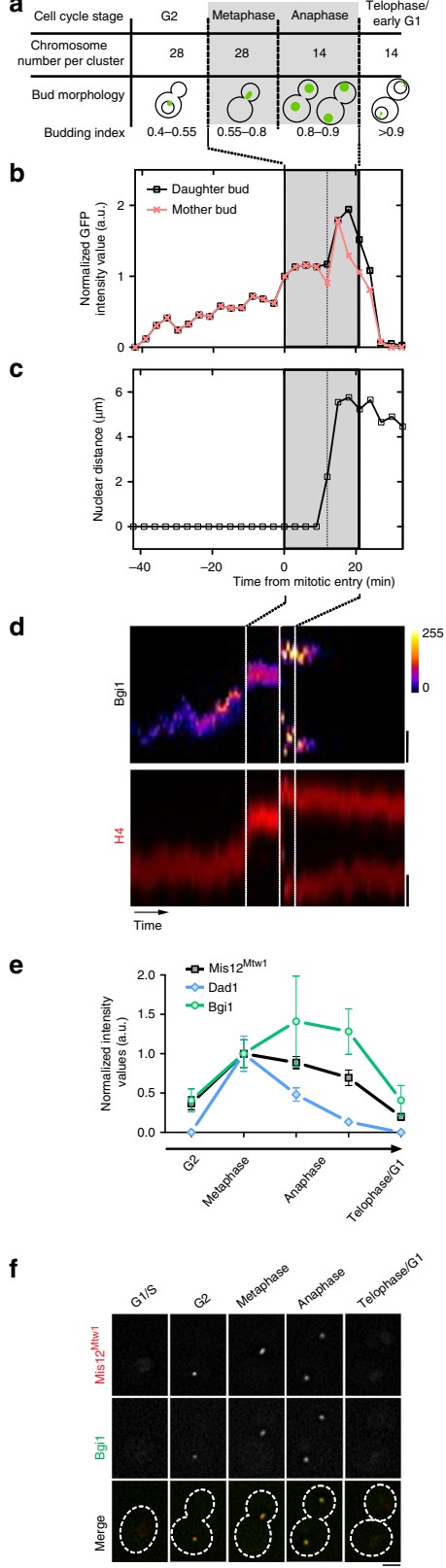

**Fig. 3 Bridgin reaches its peak concentration at the kinetochore during anaphase. a** Expected chromosome number per kinetochore cluster in the haploid-type strain H99/KN99 and spatial location of bridgin (green) with the corresponding budding index is tabulated. **b** Signal intensity measurements of Bgi1 in a single cell of the strain SHR843 (*BGI1::Bgi1-V5-GFP, H4::H4-mCherry*), at an interval of 3 min were measured and normalized to the Bgi1 signal intensity at time point 0 (M-phase entry). **c** The nuclear marker histone H4 in SHR843 was utilized to measure internuclear distance. A cell is considered to have exited anaphase when nuclear distances have reached their maxima. **d** Kymograph of the cell used for signal measurements in **b**. Time interval represented 1 min for a total of 100 min. Scale bar, 2 μm. **e** Comparison of the protein levels of Bgi1 in SHR843, and representative outer kinetochore proteins, Mis12$^{Mtw1}$ in SHR516 (*MIS12$^{MTW1}$::MIS12$^{MTW1}$-mCherry*), and Dad1 in CNVY120 (*KN99::H3p-GFP-DAD1*). Signal intensities of each kinetochore protein were quantified and normalized to the mean of their metaphase signals. The results represent the mean ± S.D. of five cells. **f** Co-localization of Bgi1 and Mis12$^{Mtw1}$ in SHR869 (*BGI1::BGI1-V5-GFP, MIS12$^{MTW1}$::MIS12$^{MTW1}$-mCherry*) at various stages of the cell cycle in an asynchronous culture. Scale bar, 3 μm. Source data are available as a Source Data file.

Dam1C subunit Dad2 (Fig. 2c). On the other hand, bridgin's kinetochore localization was entirely compromised in the Mis12$^{Mtw1}$ (Mis12C) conditional mutant (Fig. 2d). Considering that Mis12C and Ndc80C require each other for their kinetochore stability (Supplementary Fig. 3a, b), it would be difficult to distinguish their individual contributions to bridgin localization using the interdependency assay. Further, we suspect that bridgin's kinetochore localization would also be affected in Ndc80C conditional mutants. Downstream of the Mis12C–Ndc80C platform, the role of Knl1C components, Knl1$^{Spc105}$ and Sos7, on bridgin's localization at the kinetochore, was tested. Bridgin levels at the kinetochore were reduced to 69% and 41% as compared to the wild type in the conditional mutant of Knl1$^{Spc105}$ and the null mutant of *sos7*, respectively. Thus, both Knl1C components partially contributed to bridgin's kinetochore recruitment (Fig. 2e, f). Taken together, we demonstrate that the recruitment of bridgin at the kinetochore occurs downstream of Knl1C and Mis12C–Ndc80C, suggesting that bridgin has multiple binding sites within the KMN network (Fig. 2g).

**Bridgin levels at the kinetochore peak at anaphase**. To understand how bridgin dynamics is regulated during cell-cycle progression, we analyzed its signal intensities at the kinetochore. Bridgin localized to the kinetochore starting from G2 until telophase/G1 (Fig. 3a–d and Supplementary Fig. 1k). The signal intensities of bridgin at the kinetochore reached the peak immediately post-anaphase onset, attaining an average of ~150% of metaphase (Fig. 3b, d, e). The dynamic intensities of other transiently localized kinetochore proteins Mis12$^{Mtw1}$ (KMN network) and Dad1 (Dam1C) were measured. Mis12$^{Mtw1}$ localized to the kinetochore during G2 and persisted until telophase/G1, reaching the maximum signal intensity during metaphase (Fig. 3e). Dad1 (Dam1C) localized at the kinetochore exclusively postmitotic onset, reaching peak intensities at metaphase and reducing sharply to an almost undetectable level in late anaphase (Fig. 3e). Analysis of bridgin and Mis12$^{Mtw1}$ localization in an asynchronous population further validated their co-localization to the kinetochore from G2 to telophase/G1 (Fig. 3f). Having observed bridgin's enrichment at the kinetochore, its dependence on the outer kinetochore KMN proteins, kinetochore localization dynamics, and spindle-independent kinetochore recruitment, we conclude that bridgin, as an outer kinetochore protein, is closely associated with the KMN platform.

at the kinetochore. Moreover, the Ndc80C–Mis12C platform requires the inner kinetochore protein CENP-C$^{Mif2}$ for its kinetochore localization (Supplementary Fig. 3j). Observed interdependencies are summarized in Supplementary Fig. 3k.

Subsequent interdependency analyses with bridgin suggested that its localization to the kinetochore was independent of the

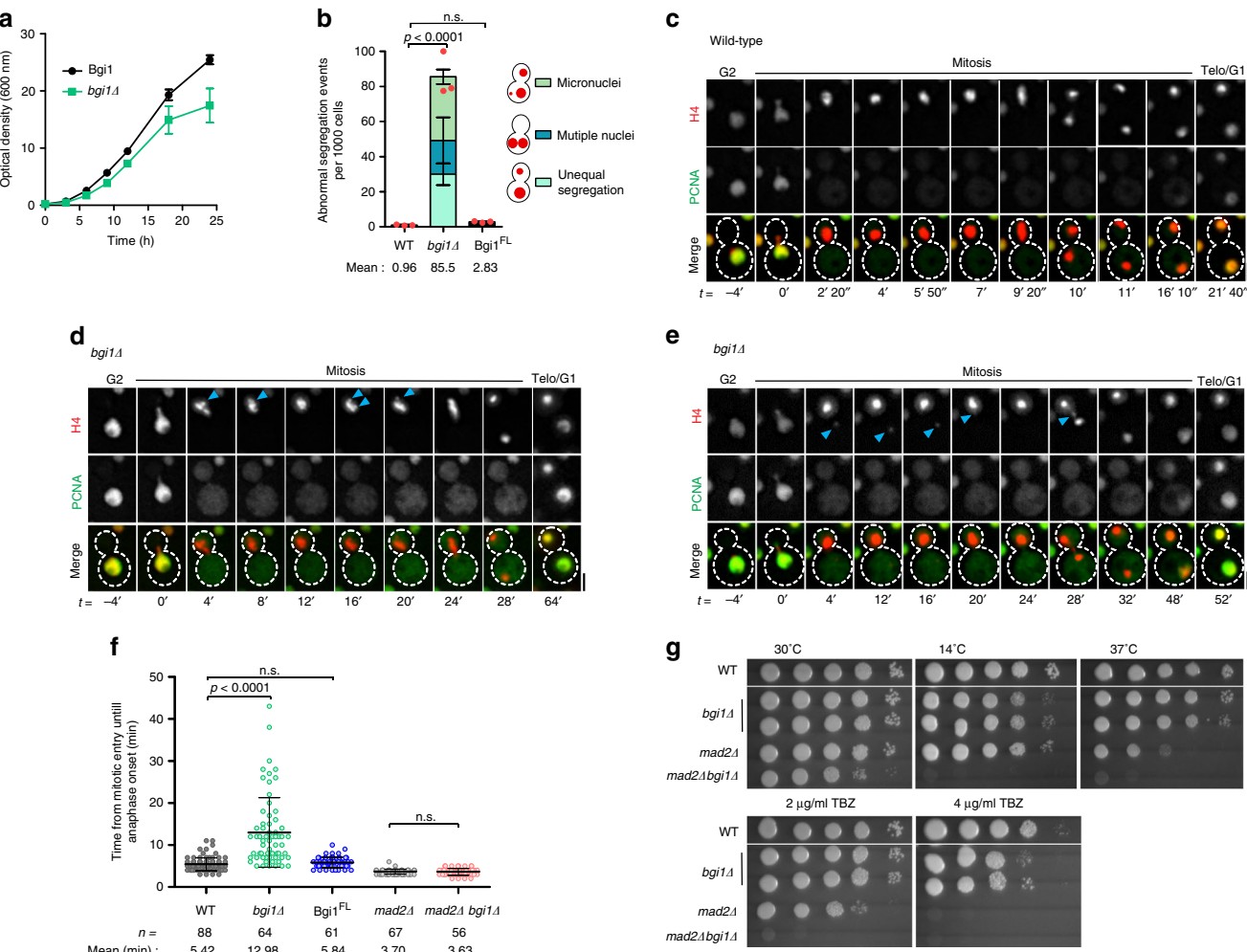

**Fig. 4 Loss of bridgin function results in an increased rate of chromosome missegregation and impaired mitotic progression. a** Growth curve of wild-type H99 and bridgin-null mutant strain SHR867 (*bgi1Δ::NEO*) at 30 °C. N = 3. **b** The rate of abnormal nuclear segregation events was measured at 30 °C using histone H4-mCherry in control CNVY121 (*BGI1*), bridgin-null mutant SHR832 (*bgi1Δ::HygB*), and the Bgi1FL-complemented SHR879 (*SHR832::3×FLAG-GFP-BGI1FL*) strains. The data represent the mean ± S.D. of three independent experiments. A red dot represents the net abnormal nuclear segregation events per 10³ cells of one experiment. One-way ANOVA test followed by Dunn's multiple comparison test was used to calculate the statistical significance of differences between the net missegregation events across strains (the *p* values show the difference compared to wild type, ns nonsignificant). **c–e** Representative time-lapse images of GFP-PCNA and histone H4-mCherry dynamics in **c** control strain SHR854 (*BGI1*) or **d**, **e** bridgin-null mutant strain SHR873 cells at 30 °C. Time measurements were made from the mitotic onset. Cell-cycle stages were scored for either PCNA localization or chromatin condensation and nuclear migration into the daughter bud. Scale bar, 2 μm. **d** Blue arrows indicate a chromosome that is separated from the compact chromatin mass prior to anaphase onset. **e** Blue arrows point to a lagging chromosome at the onset of mitosis through anaphase. **f** Time from mitotic entry to anaphase onset at 30 °C was quantified using a nuclear segregation marker, histone H4, dynamics, and plotted for wild-type control SHR854 (*BGI1*), bridgin-null mutant SHR873 (*bgi1Δ::HygB*), the Bgi1FL reintegrated strain SHR879, *mad2* null mutant SHR741 (*mad2Δ: NEO*), and the double-mutant strain SHR866 (*mad2Δ::NEO, bgi1Δ::HygB*). An internuclear distance of >1 μm was considered as the entry into anaphase. Mean ± S.D. is indicated. *n* indicates the number of live cells measured. Each dot represents a single live-cell measurement. Kruskal–Wallis one-way analysis followed by Dunn's multiple comparison test was used to calculate the statistical significance of differences (the *p* values show the difference compared to their respective controls, wild type or *mad2Δ*). **g** Serial 10-fold dilutions starting from 2 × 10⁵ cells were spotted for strains wild type (H99), *bgi1Δ* (SHR867), *mad2Δ* (SHR741), and *mad2Δ bgi1Δ* (SHR866) are shown. Source data are available as a Source Data file.

**Bridgin is essential for mitotic progression and accurate chromosome segregation**. Bridgin-null (*bgi1Δ*) strains were generated to further characterize the function of bridgin as a kinetochore protein. *bgi1Δ* cells exhibited reduced growth rates (Fig. 4a), and ~20% loss in viability as compared to its wild-type parent strain (Supplementary Fig. 4a). The *bgi1Δ* mutant cells also displayed an ~90-fold increase in the gross missegregation rate, which may account for the reduced viability of *bgi1Δ* (Fig. 4b and Supplementary Fig. 4a). These defects of *bgi1Δ* cells were complemented by the reintegration of the full-length bridgin gene, Bgi1FL, expressed under the control of its native promoter

(Fig. 4b and Supplementary Fig. 4a). We subsequently examined how *bgi1Δ* affected cell-cycle progression. For this, previously described microscopy-based markers such as histone H4 and PCNA were utilized to determine cell-cycle stages[24,53]. The results were summarized in Supplementary Fig. 4b. In the wild type, the nucleus migrated into the daughter cell as a coalesced mass at the onset of M phase, followed by its equal segregation in anaphase (Fig. 4c). On the other hand, *bgi1Δ* cells exhibited lagging chromosomes at the M phase, suggesting inaccurate kinetochore–microtubule attachments (Fig. 4d, e). The live-cell analysis revealed that while wild-type cells spent an average of 18

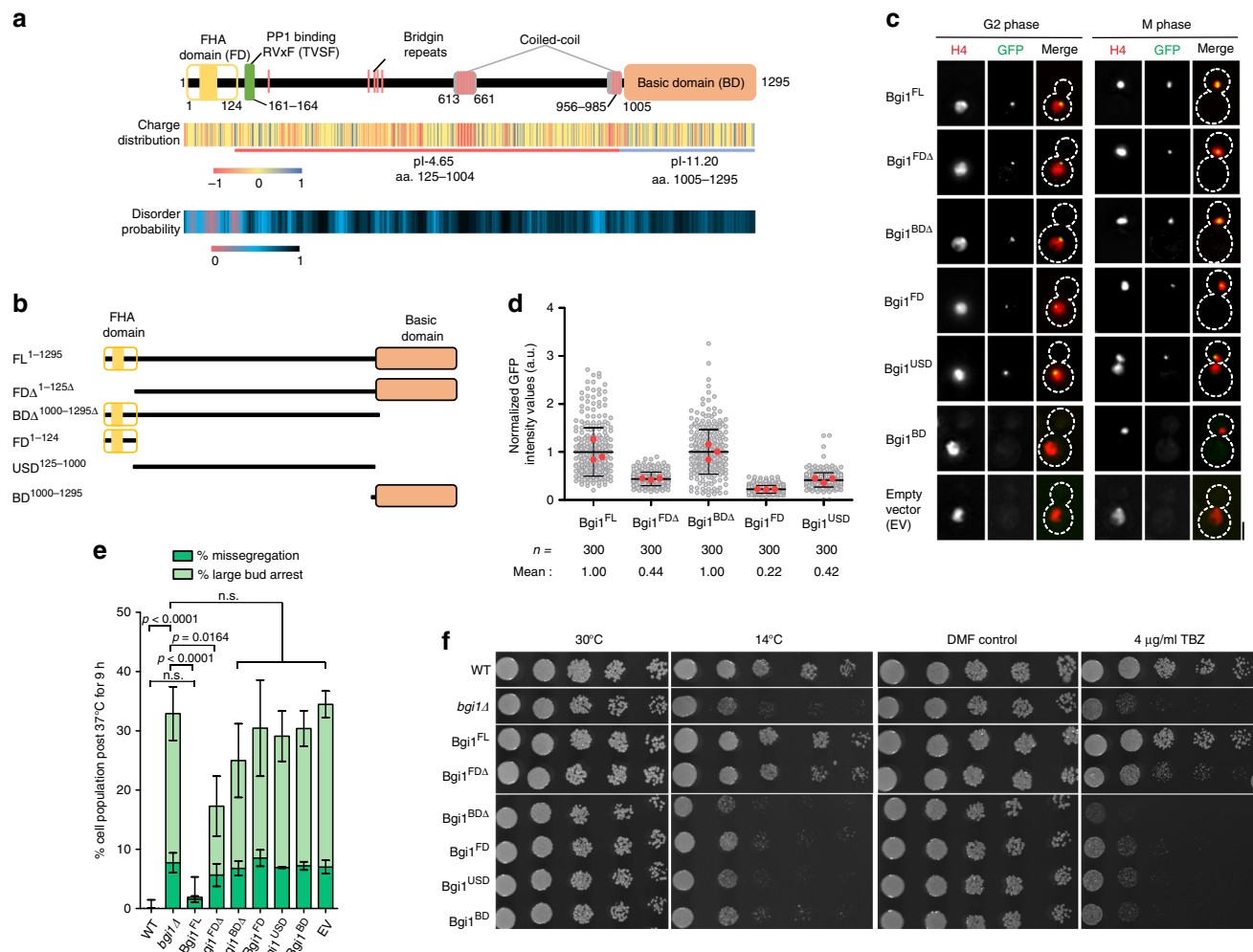

**Fig. 5 The basic domain of bridgin is dispensable for its kinetochore recruitment but not for its function. a** Schematic describing predicted features of bridgin. Charge distribution of amino acid residues was predicted with a window size of 2 using EMBOSS charge. The disorder probability of bridgin was calculated using IUPRED2A. **b** Schematic of domain deletion constructs of bridgin used in this study. Constructs were generated with a 3×FLAG-GFP tag at the amino terminus. **c** Representative cells of Bgi1^FL (SHR879), Bgi1^FDΔ (SHR913, *SHR832::3×FLAG-GFP-BGI1^FDΔ*), Bgi1^BDΔ (SHR880, *SHR832::3×FLAG-GFP-BGI1^BDΔ*), Bgi1^FD (SHR915, *SHR832::3×FLAG-GFP-BGI1^FD*), Bgi1^USD (SHR916, *SHR832::3×FLAG-GFP-BGI1^USD*), Bgi1^BD (SHR917, *SHR832::3×FLAG-GFP-BGI1^BD*), and an empty vector (EV)/FLAG-GFP control (SHR918, *SHR832::3×FLAG-GFP*) depicting localization of the mentioned GFP constructs in G2 and M phases. Nuclear localization was scored for using the chromatin marker histone H4-mCherry. Scale bar, 3 μm. **d** Signal intensities of bridgin constructs mentioned in **c** were measured at metaphase and normalized to the mean intensity of Bgi1^FL. The data represent the results of three independent experiments with 100 cells each. The red dot represents the mean of one experiment; mean ± S.D. is shown. **e** The extent of complementation of bridgin constructs mentioned in **c**, wild-type control CNVY121 (*BGI1*), and the bridgin-null mutant SHR832 was measured. Cells were grown at 30 °C until log phase and transferred to 37 °C. Indicated cell populations were measured 9 h post incubation at 37 °C. All values were normalized to wild-type control levels. Defects in nuclear segregation were measured as mentioned in Fig. 4b. The data represent the mean ± S.D. of three independent experiments. One-way ANOVA test with Tukey's multiple comparison test was used to calculate the statistical significance of differences between the net missegregation and large-bud arrest populations across strains. **f** Cells of varying numbers, $2 \times 10^4$, $2 \times 10^3$, 200, 100, and 50, of strains mentioned in **e** were spotted on plates as indicated. Dimethylformamide, DMF, solute control. Source data are available as a Source Data file.

min, *bgi1Δ* cells exhibited a delayed spending ~30 min in the M phase. This is an underrepresentation since ~10% of cells failed to exit the M phase even after >50 min (Supplementary Fig. 4c). Within the M phase, the delay occurred before anaphase onset (Fig. 4f and Supplementary Fig. 5f). Further analysis revealed that *bgi1Δ* cells were sensitive to spindle insults (Fig. 4g). Next, we tested whether *bgi1Δ*-associated defects are under the surveillance of SAC. *mad2Δ* in the background of *bgi1Δ* alleviated M-phase delay (Fig. 4f and Supplementary Fig. 4d). However, the double mutants (*mad2Δ bgi1Δ*) were conditionally synthetic lethal upon treatment of TBZ (2 μg/ml) or under conditions that compromise the spindle stability (14 and 37 °C) (Fig. 4g). Based on nuclear

segregation defects, sensitivities to spindle insults, and SAC-mediated delay in mitotic progression, we conclude that bridgin is important for accurate kinetochore–microtubule interactions required for high-fidelity chromosome segregation. These defects associated with *bgi1Δ* cells activate SAC response to correct for erroneous kinetochore–microtubule attachments.

**The basic domain of bridgin is critical for its function.** Next, we sought to understand how bridgin, being a part of the kinetochore, contributes to high-fidelity chromosome segregation. Bridgin is predicted to be 1295-amino acid (aa) long. Its N-terminal region (aa1–124) forms a forkhead-associated (FHA)

domain (a phosphopeptide recognition domain) followed by an unconventional putative PP1 docking site (aa161–164). In contrast, the remaining region of the protein is predicted to be largely unstructured (Fig. 5a). The C-terminal region (aa1005–1295) that has a pI of 11.20 was termed as the basic domain (BD). The unstructured domain (USD) was defined as the region spanning aa125–1004, which was acidic with a predicted pI of 4.65 (Fig. 5a), and contained 13 repeats of a bridgin consensus motif (Supplementary Fig. 5a and Supplementary Table 1).

Domain deletion constructs of bridgin (Supplementary Fig. 5b and Fig. 5b) wherein the various versions of bridgin are expressed under the control of its native promoter with an N-terminal 3×FLAG-GFP epitope tag. Each of these constructs was reintegrated into $bgi1\Delta$ cells expressing histone H4-mCherry to obtain strains expressing truncated bridgin proteins (Fig. 5b). Microscopic estimation of GFP signal intensities of the bridgin constructs suggested that the FHA domain (FD) and USD regions were able to localize independently of each other at the kinetochore, albeit to different extents of ~20% and ~40% of the $Bgi1^{FL}$ level, respectively (Fig. 5c, d). Localization of $Bgi1^{BD\Delta}$ at the kinetochore was not significantly different from that of $Bgi1^{FL}$ (Fig. 5c, d). Further, a lack of kinetochore localization of $Bgi1^{BD}$ suggested that BD is dispensable for the kinetochore localization of bridgin. These results suggest that bridgin through its FD and USD makes multiple contacts at the kinetochore, consistent with the observation that bridgin is recruited downstream to multiple outer kinetochore KMN protein subunits (Fig. 2g).

To define the domains necessary for bridgin's function in chromosome segregation, we scored for complementation of the $bgi1\Delta$ phenotype at 37 °C. This assay condition was used for the ease of scoring due to the enhanced phenotype of M-phase delay (Fig. 5e). In addition, cell growth assays under conditions altering microtubule dynamics (Fig. 5f), and estimation of abnormal segregation defects at 30 °C was performed (Supplementary Fig. 5c). As expected, $Bgi1^{FL}$ was able to suppress the $bgi1\Delta$ phenotype significantly (Fig. 5e, f). Partial complementation of phenotype was observed for the mutant expressing $Bgi1^{FD\Delta}$. No significant complementation was obtained for any other domain deletion constructs, including the $Bgi1^{BD\Delta}$ that localized to the kinetochore similar to $Bgi1^{FL}$ levels (Fig. 5e, f). As observed at 37 °C, comparable results were observed for the measured rate of missegregation at 30 °C, albeit weak complementation was observed for $Bgi1^{BD\Delta}$ (Supplementary Fig. 5c). Taken together, we conclude that all domains, including BD, which is not involved in kinetochore localization of bridgin, are critical for restoring full bridgin function.

**The C-terminal BD of bridgin interacts with chromatin.** The impact of bridgin on SAC activity (Supplementary Fig. 5d, e) and spindle dynamics (measured at anaphase, Supplementary Fig. 5f) was tested and found to be unaltered in $bgi1\Delta$. Thus, these factors were ruled out as possible reasons for increased missegregation associated with $bgi1\Delta$ mutants. To address the role of BD toward bridgin function, FLAG-IP for $Bgi1^{FL}$ (using 150 mM KCl and a more stringent condition of 300 mM KCl) and $Bgi1^{BD\Delta}$ (150 mM KCl) was performed and subjected to MS analysis (Fig. 6a and Supplementary Fig. 6a). A comparison of the specific interactors obtained across the $Bgi1^{FL}$ and $Bgi1^{BD\Delta}$ suggested an enrichment of chromatin-interacting proteins, including inner kinetochore proteins, in $Bgi1^{FL}$ IP as compared to $Bgi1^{BD\Delta}$ IP (Table 2 and Supplementary Data 3). In comparison, outer kinetochore KMN network proteins were identified across the $Bgi1^{FL}$ and $Bgi1^{BD\Delta}$ IPs (Table 2). Proteins of the KMN network were among the top hits in $Bgi1^{BD\Delta}$ IP (Supplementary Data 3).

Based on the observation that chromatin-interacting proteins were less recovered in $Bgi1^{BD\Delta}$ IP, we hypothesized that BD might play a role in chromatin interactions. Through co-IP experiments, histone H4 was found to associate with $Bgi1^{FL}$ (150 mM) as well as with the inner kinetochore component CENP-$C^{Mif2}$ (Supplementary Fig. 6b). On the other hand, histone H4 was greatly reduced in $Bgi1^{BD\Delta}$ IP like the outer kinetochore protein IPs of Dsn1, Spc25, and Spc34 (Dam1C) (Supplementary Fig. 6b). These results raise the possibility that the bridgin–chromatin interaction in vivo occurs directly through BD and is not a consequence of mere bridgin assembly onto the centromere-localized KMN network platform. We further tested the direct interaction of purified bridgin BD with free DNA or reconstituted nucleosomes in vitro by electrophoretic mobility shift assay (EMSA). We found that BD was able to bind to DNA (Fig. 6b) and nucleosomes of varying compositions, containing histone H3 or CENP-A (Supplementary Fig. 6c). Taking together, MS results along with in vivo- and in vitro-binding assays strongly indicated that BD of bridgin interacts with DNA/chromatin.

Although BD seems to bind DNA nonspecifically in vitro, we cannot rule out the possibility that BD may exhibit specificity to *C. neoformans* centromere DNA/chromatin. Alternatively, other factors may restrict BD's centromeric localization in vivo. If the latter possibility is true, overexpression (OE) of bridgin may not alter its localization. The OE assay entailed that chromatin-unbound, but nuclear-localized proteins would diffuse into the cytoplasm owing to increased nuclear permeability in the M phase of *C. neoformans* (Fig. 6c). This is demonstrated by GFP-PCNA, a protein with well-established roles in S phase[54]. PCNA is nuclear localized in G2, but being unbound to chromatin in the M phase, it diffuses into the cytoplasm (see below). OE of GFP-$Bgi1^{FL}$ (Supplementary 6d) resulted in its signals no longer being restricted to the kinetochores (Fig. 6d), rather signals were found to be overlapped with chromatin marked by histone H4 (Fig. 6e). Taking together, we conclude that bridgin interacts with bulk DNA/chromatin nonspecifically in vivo as well.

Further, using the OE strategy (Fig. 6e), we demonstrated that in the absence of BD, signals of $Bgi1^{BD\Delta}$-OE were restricted to a punctum, unlike $Bgi1^{FL}$-OE localization. Supporting the notion that bridgin localizes to the kinetochore through FD and USD, a single punctum for both overexpressed constructs, $Bgi1^{FD}$ and $Bgi1^{USD}$, was observed. On the other hand, OE of constructs carrying BD such as $Bgi1^{BD}$-OE, $Bgi1^{FD\Delta}$-OE, or $Bgi1^{FL}$-OE, resulted in their localization to bulk chromatin. Thus, these observations suggested that BD is necessary and sufficient to bind chromatin in vivo, and the loss of BD in the OE constructs was sufficient to restrict bridgin localization to the kinetochore. Taken together, unlike other outer kinetochore proteins, it was surprising that bridgin was able to interact with centromeric chromatin.

**Overall net positive charge of the BD is vital for bridgin function.** Toward understanding the significance of bridgin binding to chromatin, we show that loss of bridgin function neither alters the previously described chromatin marks of CpG methylation at *C. neoformans* centromeres[47] (Supplementary Fig. 6e, f) nor H3K9me2 cellular pools previously shown to be predominantly enriched across centromeres and telomeres[55] (Supplementary Fig. 6g). To summarize, we observe that BD is dispensable for kinetochore localization of bridgin, but required for its function. However, it remains unclear how BD influences bridgin's function. We hypothesize two possibilities: either BD modulates bridgin function by interacting with specific proteins at chromatin, or BD does so by interacting directly with DNA/

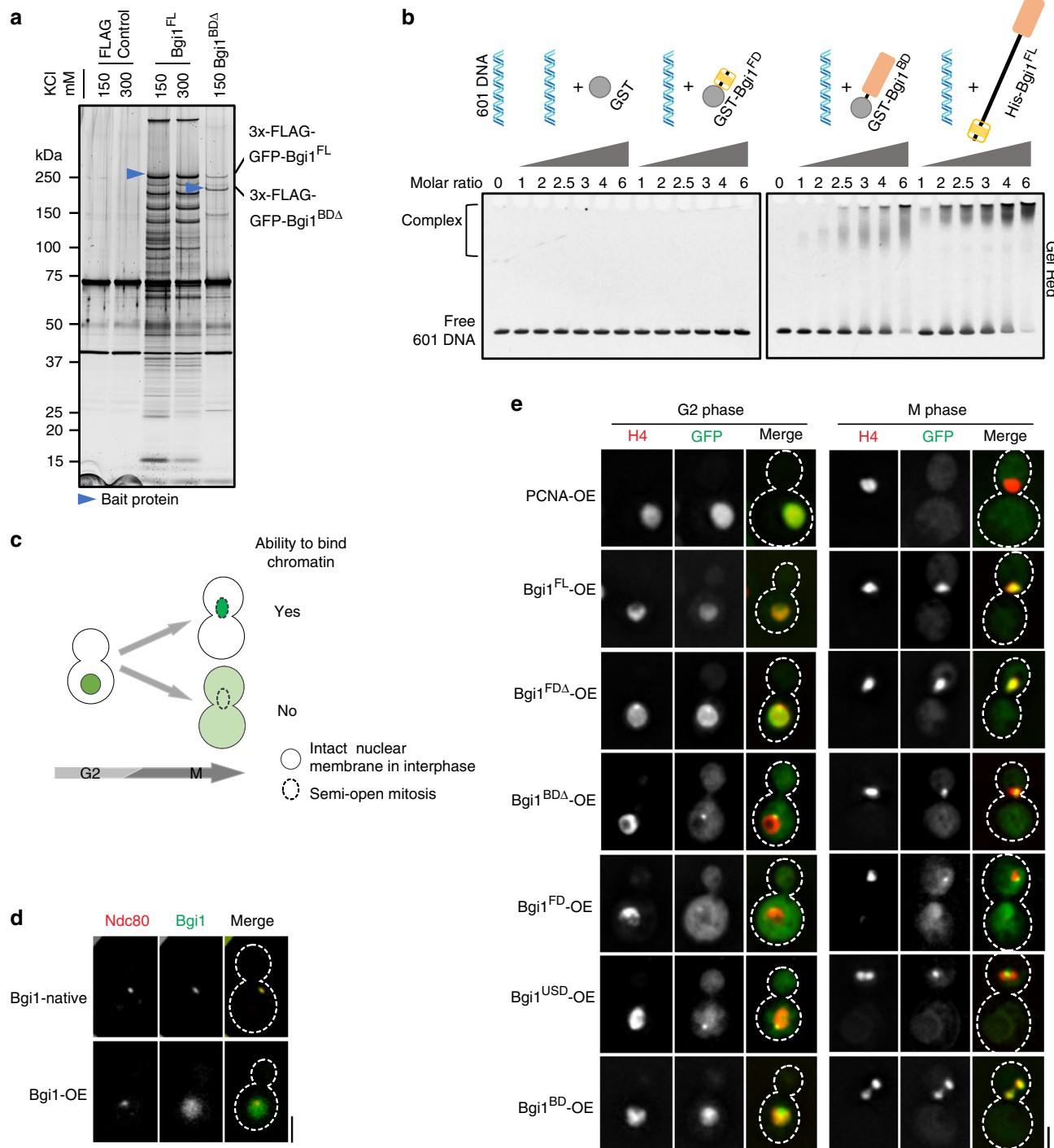

**Fig. 6 The basic domain of bridgin has a property to interact with DNA nonspecifically in vitro and in vivo. a** Proteins eluted from TBZ-treated cells after FLAG-IP of Bgi1$^{FL}$ expressing SHR879, Bgi1$^{BD\Delta}$ expressing SHR880, and a FLAG control strain SHR942 expressing Bgi1 were separated on a gradient PAGE gel and silver-stained. Blue arrows mark the bait proteins. **b** Electrophoretic mobility shift assay (EMSA) samples were separated on a PAGE gel and stained with GelRed for visualization. **c** Chromatin-bound proteins (green) colocalize with the nuclear marker histone H4-mCherry in metaphase, while free nuclear proteins diffuse into the cytoplasm following the entry into mitosis. **d** Visualization of GFP-Bgi1 when expressed under the native (SHR847, *BGI1:: BGI1-V5-GFP*) or an overexpression (OE) *GAL7* promoter (SHR858, *BGI1::GAL7p-GFP-BGI1*). Outer kinetochore protein Ndc80-mCherry was used to mark the kinetochore. **e** Visualization of bridgin OE constructs. The expression of OE constructs of Bgi1$^{FL}$ (SHR895, *SHR832::H3p-GFP-BGI1$^{FL}$*), Bgi1$^{FD\Delta}$ (SHR920, *SHR832::H3p-GFP-BGI1$^{FD\Delta}$*), Bgi1$^{BD\Delta}$ (SHR921, *SHR832::H3p-GFP-BGI1$^{BD\Delta}$*), Bgi1$^{FD}$ (SHR922, *SHR832::H3p-GFP-BGI1$^{FD}$*), Bgi1$^{USD}$ (SHR923, *SHR832::H3p-GFP-Bgi1$^{USD}$*) and Bgi1$^{BD}$ (SHR924, *SHR832::3×FLAG-GFP-BGI1$^{BD}$*), and GFP-PCNA control (SHR854, *BGI1, CNVY121::H3p-GFP-PCNA*) was visualized in G2 and M phases. Histone H4-mCherry was used to mark chromatin. Scale bar, 3 μm. Source data are available as a Source Data file.

**Table 2 List of chromatin-associated and kinetochore proteins obtained as interactors from bridgin IP's performed in Fig. 5a.**

| Complex/category | Protein | FLAG control 150 mM | | FLAG control 300 mM | | Bgi1$^{FL}$ 150 mM KCl | | Bgi1$^{FL}$ 300 mM KCl | | Bgi1$^{BD\Delta}$ 150 mM KCl | |
|---|---|---|---|---|---|---|---|---|---|---|---|
| | | C (%) | TSC | C (%) | TSC | C (%) | TSC | C (%) | TSC | C (%) | TSC |
| *Chromatin-associated proteins* | | | | | | | | | | | |
| Histone proteins | H4 | 12 | 2 | 51 | 12 | 62 | 139 | 57 | 96 | 51 | 18 |
| | H3 | – | – | 13 | 2 | 51 | 45 | 45 | 33 | 51 | 4 |
| | H2A | 6.9 | 1 | 12 | 3 | 46 | 30 | 45 | 27 | 12 | 3 |
| | H2A.Z | 6.5 | 1 | 6.5 | 2 | 47 | 13 | 49 | 13 | 6.5 | 2 |
| | H2B | – | – | 14 | 2 | 53 | 54 | 44 | 47 | 24 | 4 |
| | H1 | – | – | – | – | 51 | 18 | 18 | 4 | – | – |
| MCM complex | Mcm2 | – | – | – | – | 48 | 100 | 37 | 37 | – | – |
| | Mcm3 | – | – | – | – | 56 | 136 | 45 | 56 | – | – |
| | Mcm4 | – | – | – | – | 60 | 119 | 42 | 43 | – | – |
| | Mcm5 | – | – | – | – | 58 | 85 | 35 | 28 | – | – |
| | Mcm6 | – | – | – | – | 48 | 80 | 36 | 36 | – | – |
| | Mcm7 | – | – | – | – | 62 | 119 | 39 | 43 | – | – |
| DNA-binding proteins | Hmo1 | – | – | – | – | 36 | 27 | – | – | – | – |
| | RFA1 | – | – | – | – | 66 | 70 | 63 | 53 | – | – |
| | Ku70 | – | – | – | – | 40 | 24 | 44 | 35 | – | – |
| | Ku80 | – | – | – | – | 41 | 34 | 20 | 15 | – | – |
| *Kinetochore proteins* | | | | | | | | | | | |
| Ndc80 | Ndc80 | – | – | – | – | 24 | 14 | 17 | 10 | 9.8 | 6 |
| | Nuf2 | – | – | – | – | 30 | 14 | 31 | 12 | 27 | 14 |
| | Spc25 | – | – | – | – | 39 | 15 | 46 | 20 | 18 | 6 |
| | Spc24 | | | | | 39 | 12 | 57 | 20 | 21 | 4 |
| Knl1 | Sos7 | – | – | – | – | 37 | 12 | 35 | 16 | 2.5 | 1 |
| | Knl1$^{Spc105}$ | – | – | – | – | 21 | 32 | 13 | 22 | 24 | 34 |
| Mis12 | Nsl1 | – | – | – | – | 15 | 3 | 30 | 6 | 3.9 | 1 |
| | Nnf1 | – | – | – | – | 25 | 15 | 31 | 17 | 15 | 6 |
| | Dsn1 | – | – | – | – | 34 | 22 | 27 | 19 | 16 | 10 |
| | Mis12$^{Mtw1}$ | – | – | – | – | 37 | 11 | 53 | 21 | 14 | 5 |
| CCAN | CENP-C$^{Mif2}$ | – | – | – | – | 4.7 | 4 | – | – | – | – |
| CENP-A | CENP-A$^{Cse4}$ | – | – | – | – | 14 | 3 | 20 | 3 | – | – |
| PP1 | PP1 | – | – | – | – | 39 | 27 | 54 | 36 | 18 | 8 |
| Bridgin | Bgi1 | 14 | 17 | – | – | 67 | 853 | 67 | 1113 | 30 | 87 |

Percentage amino acid sequence coverage (C,%) and the total spectrum counts (TSC) specific to the corresponding protein are tabulated.

chromatin. To prove further, we performed a domain-swap experiment by replacing bridgin BD$^{1005-1295}$ with an amino acid stretch of similar properties (length: ~300aa, unstructured, non-specific DNA binding and a pI of ~10) found in the basic region Ki67-BD$^{2937-3256}$ of human Ki67 (Fig. 7a). No evident sequence conservation was observed between the BD of Bgi1 and the basic region of Ki67 (Supplementary Fig 7a). Ki67 was previously shown to bind nonspecifically to DNA[56] and functions as a surfactant by coating chromosomes during mitosis[57]. However, Ki67 is not reported to interact with kinetochores[58]. We confirmed that OE of Ki67$^{BD}$ localized to the entire nucleus, suggesting that it nonspecifically binds to DNA in *C. neoformans* (Supplementary Fig. 7b). Bgi1$^{FL}$, Bgi1$^{BD\Delta}$, and Bgi1$^{BD\Delta}$+Ki67$^{BD}$ were expressed under the native bridgin promoter, and each of these versions of Bgi1 was found to be localized to the kinetochore with comparable intensities when integrated into a *bgi1Δ* background strain (Fig. 7b, c). Weak complementation of bridgin function by Bgi1$^{BD\Delta}$ over *bgi1Δ* was observed (Fig. 7d). Strikingly however, Bgi1$^{BD\Delta}$+Ki67$^{BD}$ was able to complement defects observed in *bgi1Δ* to the extent that was comparable to Bgi1$^{FL}$ (Fig. 7d and Supplementary Fig. 7c). These observations were additionally validated by complementation of phenotypes associated with the Bgi1$^{BD\Delta}$ mutant by the Bgi1$^{BD\Delta}$ + Ki67$^{BD}$ chimeric protein in the spotting growth assay (Fig. 7e).

These results strongly indicated that the basic nature of BD contributes to bridgin function through its interaction with chromatin. To provide further evidence, we prepared partial MNase-digested chromatin (Supplementary Fig. 7d) to perform co-IP of various bridgin and kinetochore protein constructs with histone H4 (Fig. 7f-h). We also examined the enrichment of centromere DNA in these IP samples (Fig. 7h). These results show that Bgi1$^{FL}$, similar to the inner kinetochore linker protein CENP-C$^{Mif2}$ but unlike the KMN network component Spc25, interacts with chromatin (Fig. 7g) and enriches at the centromere under these native conditions (Fig. 7h). Bgi1$^{BD\Delta}$, on the other hand, exhibited a reduced interaction with chromatin and did not strongly enrich at the centromeres (Fig. 7g, h). Further, Bgi1$^{BD}$ was sufficient to interact with chromatin in vivo, but its binding was not restricted to centromere DNA (Fig. 7g, h). Since the Bgi1$^{BD\Delta}$+Ki67$^{BD}$ chimera behaved like Bgi1$^{FL}$, we conclude that the intrinsic basic nature of BD, rather than the specific amino acid sequence, is necessary for bridgin function (Fig. 7g, h).

**Bridgin maintains physiological levels of outer kinetochore proteins in the M phase.** Having established that bridgin connects the outer kinetochore to centromeric chromatin after being recruited by multiple receptors at the KMN network, we were keen to examine how bridgin contributes to the kinetochore integrity. Fluorescent intensities of inner kinetochore proteins, CENP-A$^{Cse4}$ and CENP-C$^{Mif2}$, and KMN network components of Mis12$^{Mtw1}$, Nuf2, and Knl1$^{Spc105}$ were analyzed at metaphase and anaphase in wild-type and *bgi1Δ* cells. No significant difference in the levels of inner kinetochore proteins was observed (Supplementary Fig. 7e, f). However, the components of the KMN

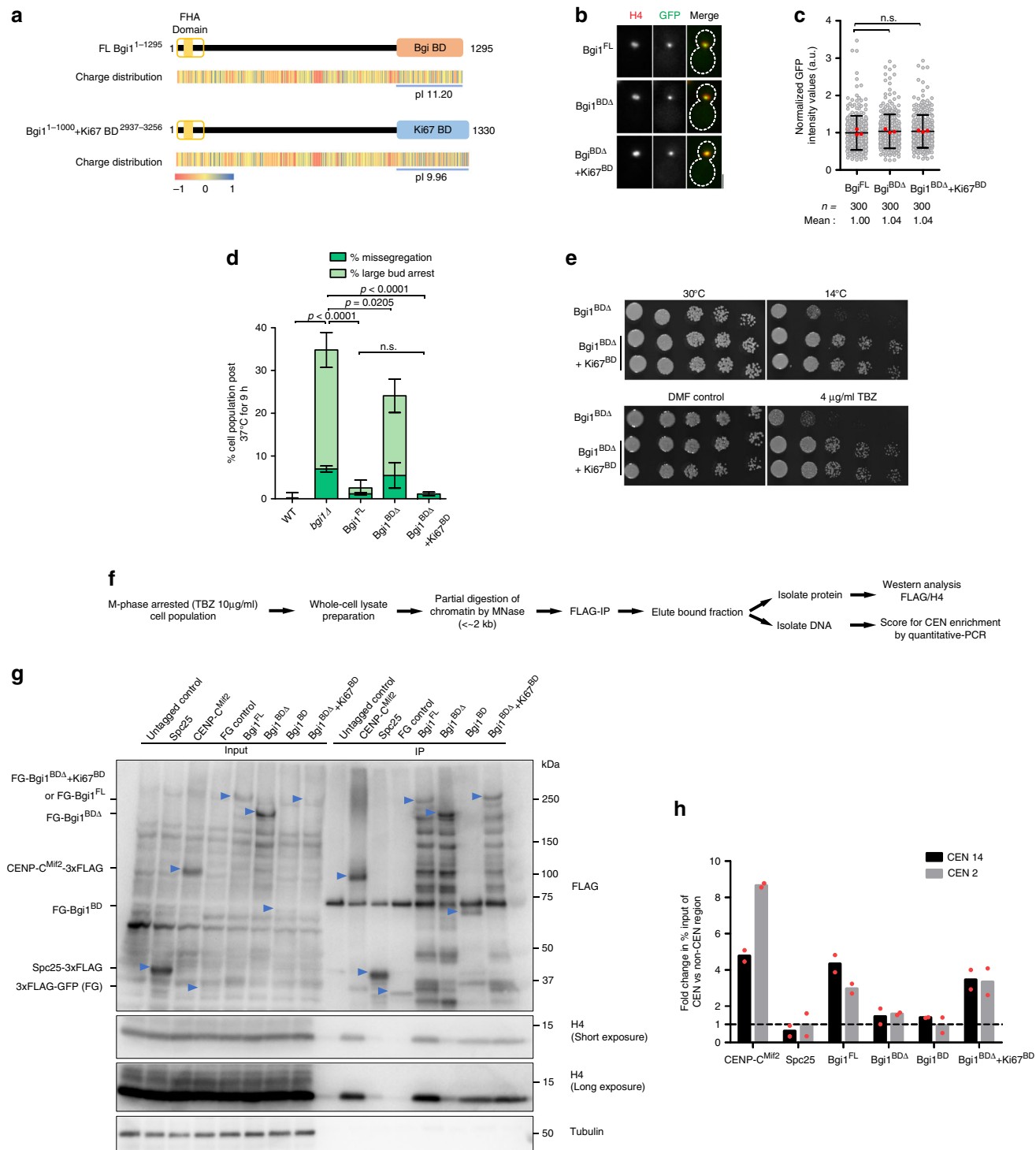

► Bait protein

network, Mis12$^{Mtw1}$ and Nuf2 in metaphase, and Mis12$^{Mtw1}$ and Knl1$^{Spc105}$ in anaphase were reduced by a small but significant extent (Supplementary Fig. 7g–i).

In summary, we identified a kinetochore protein bridgin, which derives its ability to coordinate accurate chromosome segregation in a unique way by interacting with the outer kinetochore and centromeric chromatin. We propose that bridgin connects the outer kinetochore with centromeric chromatin following its recruitment to the outer kinetochore KMN network that also restricts bridgin's binding to centromeric chromatin.

## Discussion

Our in silico analysis suggested that most inner kinetochore CCAN proteins were lost multiple independent times in the fungal phylum of Basidiomycota. To understand the unusual kinetochore composition that might have compensated for the loss of CCAN proteins, we analyzed the kinetochore interactome in the model basidiomycete, an important human fungal pathogen *C. neoformans*. In this study, we identified an outer kinetochore protein bridgin, along with all other evolutionarily conserved kinetochore proteins that are predicted to be present in

**Fig. 7 Interaction of bridgin with centromeric chromatin is established by the basic domain enriched with positively charged residues. a** Schematic of the bridgin constructs wherein its basic domain was replaced with the basic DNA-binding domain from human Ki67. **b** Representative micrographs of Bgi1[FL] expressing SHR879, Bgi1[BDΔ] expressing SHR880, and the domain-swap chimera Bgi1[BDΔ] + Ki67[BD] expressing SHR926 (*SHR832::3×-FLAG-GFP-Bgi1[BDΔ]+Ki67[BD]*). Scale bar, 3 μm. **c** Signal intensities of bridgin constructs mentioned in **b** were measured at metaphase and normalized to the mean intensity of Bgi1[FL]. The data represent the results of three independent experiments with 100 cells each. The red dot represents the mean of one experiment; mean ± S.D. is shown. Kruskal–Wallis one-way analysis followed by Dunn's multiple comparison test was used to calculate the statistical significance of differences (the *p* values show the difference compared to Bgi1[FL]). **d** The extent of complementation of bridgin constructs mentioned in **c**, wild-type control CNVY121, and the bridgin-null mutant SHR832 was measured. Indicated cell populations were measured 9 h post incubation at 37°C. All values were normalized to wild-type control levels. Defects in nuclear segregation were measured as mentioned in Fig. 4b. The data represent the mean ± S.D. of three independent experiments. One-way ANOVA test with Tukey's multiple comparison test was used to calculate the statistical significance of differences between the net missegregation and large-bud arrest population across strains. **e** Cells of varying numbers, $2 \times 10^4$, $2 \times 10^3$, 200, 100, and 50, of Bgi1[BDΔ] expressing SHR880 and the domain-swap chimera Bgi1[BDΔ]+Ki67[BD] expressing SHR926 strains, were spotted on mentioned media plates. **f** Experimental design to determine Bgi1-BD interaction with chromatin in vivo by MNase IP. **g** FLAG immunoprecipitation of kinetochore proteins. 3×FLAG-tagged proteins of Spc25 (SHR861,*SPC25::SPC25-3×FLAG*), CENP-C[Mif2] (SHR896), FLAG-GFP (FG) control (SHR918), Bgi1[FL] (SHR879), Bgi1[BDΔ] (SHR880), Bgi1[BD] (SHR917), the domain-swap chimera Bgi1[BDΔ]+Ki67[BD] (SHR926), and untagged wild-type control (H99) were pulled down and levels of interacting histone H4 were determined. Bait proteins are marked by blue arrows. **h** Relative enrichment of FLAG-tagged proteins described in **g** at centromeres following the MNase IP. qPCR was performed using CEN2 and CEN14 and nonentromeric primer sets described in Fig. 2a followed by normalization to noncentromeric controls. The data represent the mean ± S.D. of two independent experiments. Source data are available as a Source Data file.

*C. neoformans*. Our bioinformatic predictions and experiments converge on the fact that the CENP-C[Mif2] pathway is the single known linker pathway connecting centromeric chromatin to the outer kinetochore in *C. neoformans*, reminiscent of *Drosophila melanogaster* and *Caenorhabditis elegans* like kinetochores[26,59–62]. We demonstrate that bridgin through its FD and unstructured domain (USD) is recruited by the outer kinetochore and interacts with centromeric chromatin via its basic domain (BD) (Fig. 8a). Strikingly, unlike previously defined linker proteins CENP-C[Mif2] [38,39,63], CENP-T[Cnn1] [14,28,64], or CENP-U[Ame1] [39,40], that require other inner kinetochore components for their recruitment[1], bridgin is recruited by the outer kinetochore KMN network. Thus, the identification and characterization of bridgin in this study reveal an alternate pathway originating at the KMN network to connect the outer kinetochore with centromeric chromatin (Fig. 8a).

We describe that bridgin BD alone cannot specifically localize to centromeric chromatin. In addition, we demonstrate that BD's binding to centromeric chromatin is reliant on its positively charged residues rather than the primary amino acid sequence. Based on these results, we hypothesize that the binding of BD to centromeric chromatin is a consequence of specific kinetochore recruitment of bridgin through its FD and USD (Fig. 8a). Further, optimum levels of bridgin help restrict its kinetochore localization. It is noteworthy that the kinetochore localization of bridgin is dependent on multiple KMN network components, Knl1[Spc105] and Sos7 of the Knl1C, and the Mis12C–Ndc80C platform. Coincidentally, Bgi1[FD] and Bgi1[USD] independently exhibit kinetochore localization. These observations prompt us to speculate that each of the individual domains, FD and USD, may regulate interactions with a specific KMN component and their synergistic activity is important for maintaining cellular levels of bridgin at kinetochores. We are unable to determine the independent contribution of Ndc80C and Mis12C, due to their interdependency, on bridgin recruitment. Reciprocally, the reduced levels of KMN components at the M phase in *bgi1Δ* cells may be a consequence of bridgin's influence on the maintenance of the KMN components. These questions and findings necessitate further examination in the future.

Dsn1 autoinhibition[34,35,37], which diminishes the Mis12C–CENP-C interaction, must be overcome by multiple kinetochore linker pathways. The essentiality of linker pathways to overcome Dsn1 autoinhibition is species-specific. While the Dsn1 autoinhibitory domain is absent in *D. melanogaster*[60,65], that possesses a single kinetochore linker pathway CENP-C, *C.*

*neoformans* retains the domain (Supplementary Fig. 8a). Although a recent study suggests Nnf1 to be the Dsn1 homolog in *D. melanogaster*, we were unable to identify the presence of the Dsn1 autoinhibitory domain in the suggested homolog[26]. Bridgin levels at the kinetochore reach the peak at anaphase, a stage of the cell cycle when Aurora B[Ipl1]-mediated Dsn1 phosphorylation is suggested to be countered by the phosphatase activity of PP1[34,35] (Fig. 8a). The sharp reduction of Aurora B[Ipl1] localization at anaphase kinetochores[53] and the conservation of the amino terminus of CENP-C[Mif2] (Supplementary Fig. 8b) are observed in *C. neoformans*. Taken together, we propose that the kinetochore architecture alters during the metaphase–anaphase transition and that the bridgin pathway functions to reinforce/stabilize the outer kinetochore. It will be intriguing to test whether the presence of Dsn1 autoinhibition can provide a constraint driving evolution/maintenance of multiple outer kinetochore linker pathways required for outer kinetochore reinforcement in organisms with monocentric chromosomes.

Outer kinetochore proteins are found to be more conserved than their inner kinetochore counterparts, including linker proteins across eukaryotes[25,26]. Thus, additional connections mediated by factors recruited by KMN proteins, like the bridgin pathway, may provide cells with an effective alternative toward outer kinetochore reinforcement. Bridgin homologs are identified across all subphyla in Basidiomycota (Fig. 8b). Strikingly, an inability to identify bridgin homologs in specific orders correlate with the presence of multiple known linker pathways (Fig. 8b). It would be worth investigating whether the presence of the bridgin pathway allowed more flexibility in the retention/loss of specific linker pathways in basidiomycetes. Genome sequencing of many basidiomycetes would help address the correlation. Further, it would be intriguing to recognize the contribution of the multiple linker pathways in organisms like *Ustilago maydis*, which retained CENP-T, CENP-C in addition to bridgin.

Bridgin homologs could be identified in the basal ascomycetous fungi of the class Pneumocystidales, such as *Pneumocystis jirovecii* (causative organism of pneumonia), and Taphrinales. Moreover, the existence of bridgin-like proteins outside fungi may suggest an ancient origin of this family of kinetochore proteins containing an FD (Supplementary Fig. 8c and Supplementary Data 4). Ki67, a component of the mitotic chromosome periphery[57,66] but not reported to be associated with the kinetochore[58], is the identified bridgin homolog in metazoa (Supplementary Fig. 8c). This raises interesting questions regarding the functional divergence and neofunctionalization of bridgin

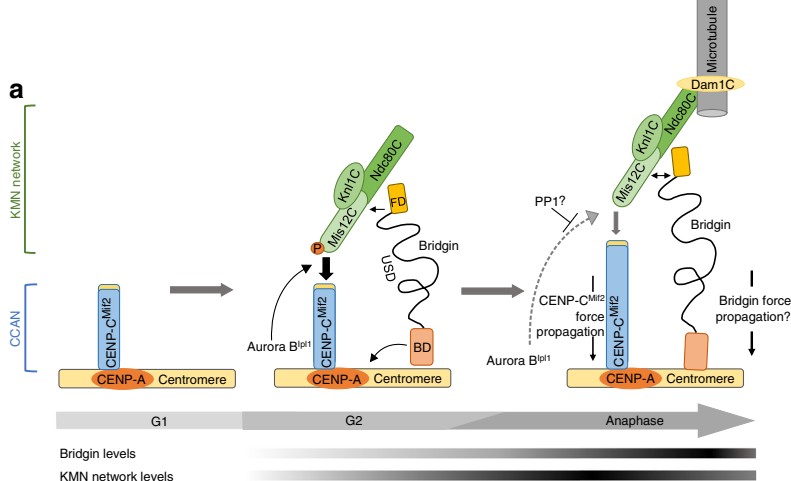

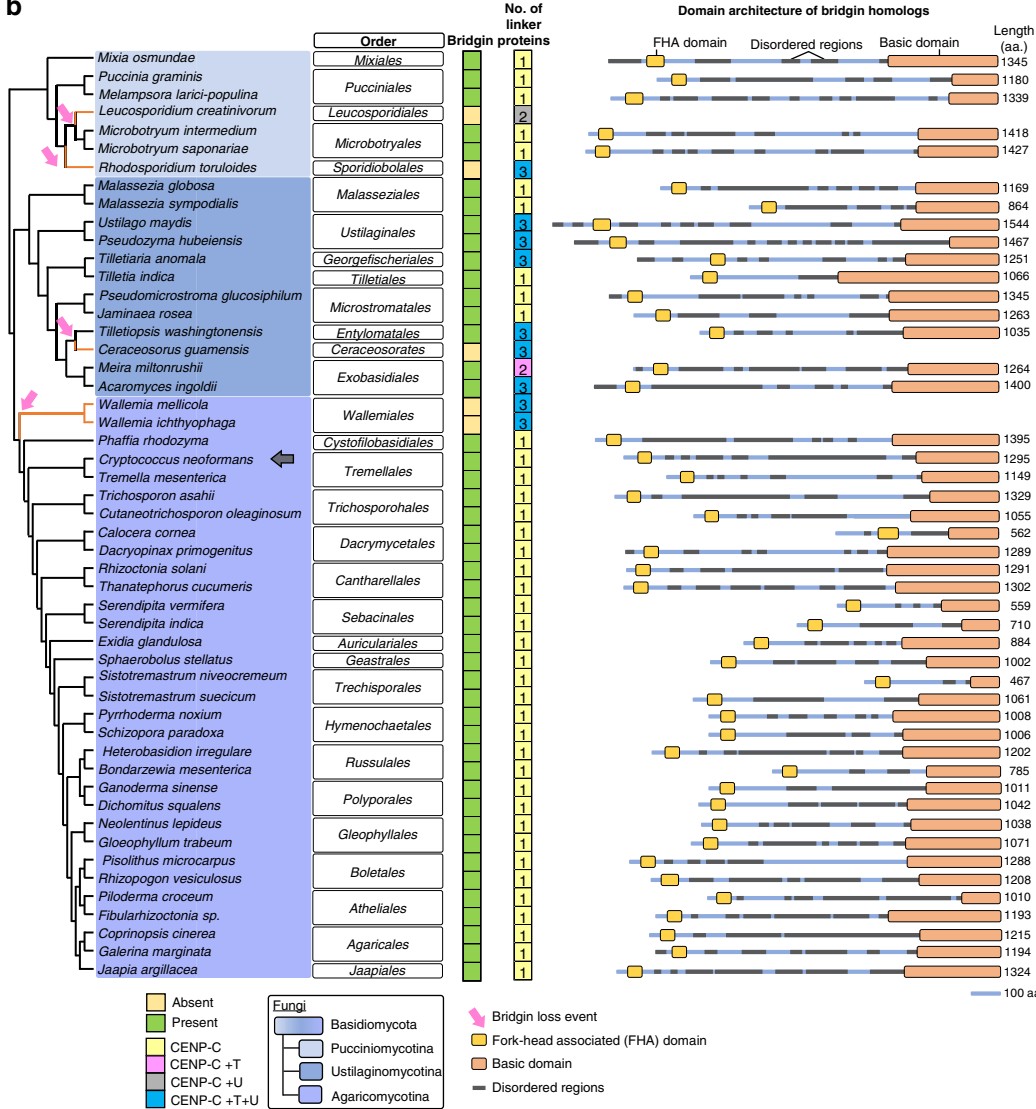

homologs invertebrates. Future experiments will reveal how bridgin is recruited onto kinetochore, how its chromatin binding is regulated, and to what extent bridgin can bear the load at the kinetochore.

There are limited experimental studies on kinetochores in nontraditional model eukaryotes[25,67] and it, therefore, remains unclear whether there are lineage-specific kinetochore components because proteins such as bridgin cannot be identified by means of bioinformatics using known kinetochore proteins as a query. Thus, the identification of such kinetochore components emphasizes the significance of studying nonconventional systems.

**Fig. 8 Bridgin, a protein conserved across most basidiomycete species that lost one or more linker proteins, can bridge the outer kinetochore KMN network and centromeric chromatin. a** A model describing bridgin as a kinetochore protein connecting the outer KMN network, through its FD and USD, and centromeric DNA directly via its BD. Restricted interaction of bridgin with centromeric chromatin in G2/M cells is a possible consequence of its outer kinetochore-specific recruitment. Increased bridgin localization is observed in anaphase. **b** Identification of bridgin homologs across Basidiomycota. *Left*, the presence (green box) or absence (yellow box) of a bridgin homolog is represented. The number of identified linker pathways has been mentioned and color-coded to represent the linker pathway(s) present. Bridgin loss events in basidiomycete lineages are represented by orange lines in the cladogram. *Right*, the domain architecture of identified bridgin homologs in Basidiomycota.

## Methods

**Homolog detection**. All searches were carried out in the NCBI nonredundant protein database and the UniProtKB database as available in July 2019. Searches for kinetochore homologs were initially carried out using iterative HMMER[68] jackhammer searches ($E$ value ≤ $10^{-3}$) with Pfam models for the mentioned kinetochore proteins. When available, models of both yeast and metazoan kinetochore homologs were considered. Obtained hits were validated by performing reciprocal HMMER searches. The secondary structure of obtained hits was validated using Jpred4 and tertiary structure prediction using HHpred[69] and/or Phyre2[70]. Protein sequences that were unable to produce hits upon reciprocal searches or failed to conform to expected secondary and tertiary structures were discarded. Further searches were performed with the same criteria using identified homologs phylogenetically closest to the species in question. Species considered in the study are mentioned in Supplementary Data 1, when homologs were not identified from a specific strain, an obtained homolog from another strain of the same species was considered. Obtained hits when possible were validated with the identified homologs from *C. neoformans*. Known kinetochore homologs from *S. cerevisiae*, *S. pombe*, *D. melanogaster*, and *H. sapiens* were used to draw the matrix of kinetochore homologs.

Toward identifying homologs of bridgin, the conserved FD was taken as the bait for subsequent iterative HMMER jackhammer searches. Obtained hits were further screened for overall protein architecture (amino-terminus FD, an unstructured central region, and a basic carboxy-terminus (Fig. 8b and Supplementary Fig. 8c). The probability of protein disorder was predicted using IUPred2A[71], and pI of the amino acid residues was predicted using ProtParam[72]. Using published multigene and genome-scale phylogenetic data from The Fungal Kingdom[48], JGI MycoCosm[73], Interactive Tree of Life (iTOL) v4[74], and Wang et al.[75], the cladograms were drawn showing the relationship among the considered species.

Sos7 and bridgin alignments were generated using T-COFFEE v11.0 and visualized using JalView v2.11.1.0.

**Yeast strains and plasmids**. A list of strains and plasmids used in the study can be found in Supplementary Table 2 and Supplementary Table 3, respectively. Primers used to generate the constructs are mentioned in Supplementary Table 4. Conditional kinetochore mutant strains were grown on 1% yeast extract, 2% peptone, and 2% galactose (YPG). All other strains were grown in 1% yeast extract, 2% peptone, and 2% dextrose (YPD) at 30 °C, 180 rpm, unless mentioned otherwise. Strains were maintained on YPD/YPG solidified with 2% agar and stored at 4 °C or −80 °C in 15% glycerol. Yeast strains are based on the haploid-type strain H99 or KN99 and generated by the standard procedure as previously described[24]. In brief, created native tagging and the *GAL7* promoter[76] replacement cassettes were excised from the plasmid construct, appropriate restriction enzymes linearized OE cassettes, and deletion cassettes were generated by overlap polymerase chain reaction (PCR) and transformed into *C. neoformans* strains of appropriate background by biolistic transformation[77]. Transformed cells were selected on drug selection in YPG for the *GAL7* promoter replacement strains to generate conditional mutants and YPD for all other strains.

**Protein affinity purification and native ChIP**. An overnight culture was inoculated at 0.1 OD$_{600}$ into fresh YPD, grown until ~0.7 OD$_{600}$, and treated with 10 μg/ml of TBZ for 3 h. Cells were harvested, washed once in water followed by one wash with binding buffer BB150 (25 mM HEPES, pH 8.0, 2 mM MgCl$_2$, 0.1 mM EDTA, 0.5 mM EGTA, 0.1% NP-40, 150 mM KCl, 1× complete EDTA-free protease inhibitor (Roche), 1× PhosStop (Roche), and 15% glycerol). Cells were resuspended in binding buffer (100 OD$_{600}$/ml). Bead beating was used to lyse the cell suspension until ~80% cell lysis was obtained. Lysates were centrifuged at 20,000$g$ for 20 min, and the supernatant was collected. The extracted cell lysate was incubated with anti-FLAG M2 antibodies (Sigma) conjugated to Dynabeads™ M-280 sheep anti-mouse IgG (ThermoFisher Scientific) for 2 h at 4 °C, under constant rotation. Unbound proteins were collected as flow-through, and proteins bound to antibody-conjugated beads were washed five times with BB150 w/o glycerol. Invert mixing was followed during each wash. Bound proteins were eluted in BB150 w/o glycerol + 200 μg/ml of 3× FLAG peptide (Sigma). Two elutes of ½ volume each of the initial bead volumes was taken and pooled.

In all, 1 μg of Anti-FLAG M2 antibody was conjugated to 10 μl of Dynabeads™ M-280 sheep anti-mouse IgG (ThermoFisher Scientific) in 1× phosphate-buffered saline (PBS), pH 7.4, and incubated for 1 h at room temperature (RT). It was washed twice with 1× PBS and resuspended in PBS. These anti-FLAG-conjugated beads were used for the lysate prepared from 100 OD$_{600}$ culture. Affinity purification samples that were processed subsequently for MS were started from a 2.25L culture, yielding ~4500 OD$_{600}$ cells. In all, 300 mM KCl, where mentioned, was used throughout the affinity purification experiment as part of the binding buffer yielding BB300.

For GFP affinity purification, GFP-Trap agarose beads (ChromoTek) were used. Bound proteins were eluted by boiling the beads for 10 min in 1× sample loading buffer (50 mM Tris-HCl, pH 6.8, 2% sodium dodecyl sulphate (SDS), 0.05% bromophenol blue, 10% glycerol, and 5% 2-mercaptoethanol) and the supernatant was collected. Other steps of the affinity purification protocol were kept the same as mentioned above.

For MNase co-IP and native ChIP, lysates prepared after bead beating were subject to centrifugation at 3300$g$ for 10 min. Following this, the supernatant was treated with MNase at RT to obtain digested chromatin (~<2 Kb in length). Post digestion, the solution was centrifuged at 20,000$g$ for 20 min. Subsequent steps followed were as described for the affinity purifications. For MNase native ChIP, DNA from the elute and input sample was extracted using GeneJET PCR purification kit (ThermoFisher Scientific). qPCR for input and IP was set up using centromere primers CN1 (CEN14)-5′-CCATCCAGTTCTTGCTTGAG-3′ 5′-GCA AGGAATGTGTTGTCTGG-3′ and CN3 (CEN2)-5′-CAGACCCTTCCTTCAG CCG-3′ 5′-TGGCAAGGAGTCGTCAGCG-3′ and noncentromeric primer set NC3 5′-GATCAAGTATAGGCGAAGG-3′ 5′-ATCTCTTATTCCCACTTCTACTC-3′ located ~825 kb away from the centromere on chromosome 1. The reactions were supplemented with SYBR green (BioRad). Data were obtained using StepOne v2.3 (Applied Biosystems). Values were normalized to percent input observed in centromeric versus noncentromeric regions and plotted using GraphPad Prism.

**Immunoblot analysis**. For whole-cell lysates, 3 OD$_{600}$ cells were harvested and resuspended in 15% TCA overnight. About 500-μl 0.5-mm glass beads were added, and samples were vortexed for a total time of 15min, with intermittent cooling on ice. They were centrifuged at 15,000$g$ for 10 min, and the obtained pellet was washed twice with 100% acetone, air-dried, and resuspended in 1× sample loading buffer and boiled for 10 min. Samples were separated on a sodium dodecyl sulphate polyacrylamide gel electrophoresis (SDS-PAGE) and transferred to Immobilon-P (Merck).

For Supplementary Figs. 1c, 2d, e, and 6g, primary antibody and secondary antibody dilutions were made in skim milk. Proteins bound by antibodies were detected with Clarity western ECL (BioRad) and visualized with Versadoc (BioRad). For Fig. 7g and Supplementary Fig. 1a, 2b, 6a, b, d primary and secondary antibody dilution were prepared in Signal Enhancer Hikari (Nacalai tesque). ChemiDoc Touch (BioRad) was used to visualize proteins reacting with antibody in the presence of the substrate ECL Prime (GE Healthcare). ImageJ[78,79] and Image lab v6.0.0 (BioRad) was used to visualize and process images. Antibodies used are tabulated in Supplementary Table 5. Uncropped blots are presented in the Source Data File.

**Mass spectrometry**. Affinity-purified samples were separated on an SDS-PAGE followed by silver staining. Silver-stained gels were visualized using an Epson GT-X980 scanner equipped with Epson Scan (Seiko Epson) v5.1.1.0. Isolated samples from the stained gel were Trypsin digested. Samples were subject to nano LC–MS–MS as described previously[80]. Briefly, after the gel slices were treated with DTT and iodoacetamide, in-gel digestion with 10 μg/ml modified trypsin (Sequencing grade, Promega) was performed at 37 °C for 16 h. The digested peptides were extracted with 1% trifluoroacetic acid and 50% acetonitrile, dried under a vacuum, and dissolved in 2% acetonitrile and 0.1% formic acid. The peptide mixtures were then fractionated by C18 reverse-phase chromatography (3 μm, ID 0.075 × 150 mm, CERI, ADVANCE UHPLC, AMR Inc.) at a flow rate of 300 nL/min with a linear gradient of 5–35% solvent B over 20 min. The compositions of solvents A and B were 0.1% TFA in water and 100% acetonitrile, respectively. The data-dependent acquisition was performed by a hybrid linear ion trap mass spectrometer (LTQ Orbitrap Velos Pro, ThermoFisher Scientific) with Advanced Captive Spray SOURCE (AMR Inc.). The mass spectrometer was programmed to carry out 11 successive scans. The first scan was performed as a full-MS scan over the range 350–1800 $m/z$ using Oubitrap analyzer at a resolution of 60,000. The second-to- eleven scan events were detected by ion trap analyzer with automatic data-dependent MS/MS scans of the top 10 most abundant ion obtained in the first scan. MS/MS spectra were obtained by setting relative collision energy of 35% CID and exclusion time of 20 s for molecules of the same $m/z$ value range.

Using the MASCOT ver2.6.2 search engine in Proteome Discoverer 2.1.1.21 and 2.2.0.388 (ThermoFisher Scientific), the obtained spectra peaks were assigned using the UniProt proteome database for *C. neoformans* H99 database (ID: UP000010091 20171201downloaded (7340 sequences)). Fragment tolerance 0.80 Da (Monoisotropic), parent tolerance 10 PPM (Monoisotropic), fixed modification of +57 on C (Carbamidomethyl), variable modification of +16 on M (oxidation) and +42 on peptide amino terminus (Acetyl), and allowing for a maximum of 2 missed cleavages for CENP-C$^{Mif2}$, Dsn1, and Spc25 and 3 missed cleavages for bridgin samples. Scaffold (version Scaffold_4.10.0, Proteome Software Inc., Portland, OR) was used to validate MS/MS-based peptide and protein identifications. Peptide identifications were accepted if they exceeded specific database search engine thresholds. Protein identifications were accepted if they contained at least two identified peptides. Proteins that contained similar peptides and could not be differentiated based on MS/MS analysis alone were grouped to satisfy the principles of parsimony. Proteins sharing significant peptide evidence were grouped into clusters. Identified protein hits from CENP-C$^{Mif2}$, Dsn1, Spc25, and their untagged controls can be found in Supplementary Data 2. To relatively quantitate protein abundance obtained within each of the experiments, Bgi1$^{FL}$ 150 mM, Bgi1$^{FL}$ 300 mM, and Bgi1$^{BDΔ}$ 150 mM, emPAI[81] (exponentially modified protein abundance index) values were determined using Scaffold 4.10.0 (Proteome Software). The higher the emPAI score, the more abundant the protein is in the mixture. Supplementary Data 3 summarizes the identified interacting protein hits from Bgi1$^{FL}$, Bgi1$^{BDΔ}$, and their untagged control IPs.

**Cross-linked ChIP and quantitative real-time PCR.** ChIP assays were performed with some modification of previously described protocols[82,83]. In brief, 100 ml of Bgi1-GFP strain was grown until ~1 OD$_{600}$. Cross-linking was performed for 20 min using formaldehyde to a final concentration of 1% and incubated at RT with intermittent mixing. The reaction was quenched by the addition of 2.5 M glycine and further incubated for 5 min. Fixed cells were harvested by centrifugation and resuspended in 9.5 ml of deionized water, followed by the addition of 0.5 ml of 2-mercaptoethanol and incubated at 30 °C for 60 min at 180 rpm. Cells were pelleted and resuspended in 10 ml of spheroplasting buffer (1 M sorbitol, 0.1 M sodium citrate, and 0.01 M EDTA) containing 40 mg of lysing enzyme from *Trichoderma harzianum* (Sigma). Spheroplasts were washed once with 15 ml each of the following buffers: (1) 1× PBS, pH 7.4, (2) Buffer I (0.25% Triton X-100, 10 mM EDTA, 0.5 mM EGTA, and 10 mM Na-HEPES, pH 6.5), and (3) Buffer II (200 mM NaCl, 1 mM EDTA, 0.5 mM EGTA, and 10 mM Na-HEPES, pH 6.5). Following which the spheroplasts were resuspended in 1 ml of extraction buffer (50 mM HEPES, pH 7.4, 1% Triton X-100, 140 mM NaCl, 0.1% Na-Deoxycholate, and 1 mM EDTA) and sonicated to shear chromatin using a Bioruptor (Diagenode) for 30 cycles of 30-s on and 30-s off bursts at high-intensity setting. Sheared chromatin was isolated in the supernatant fraction after centrifugation for 15 min at 15,000g. Average chromatin fragment sizes ranged from 200 to 500 bp. In all, 100 μl, 1/10th the volume, of the chromatin fraction, was kept for input DNA preparation, the remaining chromatin volume was divided into two halves of 450 μl each for (+) antibody and (−) antibody. For (+) antibody, 20 μl of GFP-Trap agarose beads (ChromoTek) and 20 μl of blocked agarose beads (ChromoTek) was added to (−). The tubes were incubated for 8 h to overnight on a rotator at 4 °C. Following which the supernatant was isolated as flow-through, and the beads were washed twice with low-salt buffer, twice with the high-salt buffer, once with LiCl buffer, and twice with TE. Bound chromatin was eluted in two 250-μl elution using elution buffer. All three fractions (SM, (+Ab), and (−Ab)) were de-cross-linked (mixed with 20 μl of 5 M NaCl and incubated at 65 °C for 8 h to overnight), Proteinase K treated (10 μl of 0.5 M EDTA, 20 μl of 1 M Tris-HCl, pH 6.8, 40 mg of Proteinase K was added to the solution and incubated for up to 2 h at 45 °C). DNA was isolated using phenol:chloroform extraction followed by ethanol precipitation. Isolated DNA was air-dried and dissolved in 25 μl of deionized water containing 25 μg/ml of RNase (Sigma).

All three samples (SM, (+), and (−) antibody) were subject to real-time quantitative PCR. The reaction mixture was set up using the iTaq™ universal SYBR green Supermix (BioRad) with 1 μl of the undiluted (+Ab), (−Ab) DNA samples, and SM (diluted 1:50). qPCR primers described above for the native ChIP were used. Data were obtained using CFX manager v3.1 (BioRad). Enrichment at centromeric regions was normalized to a control noncentromeric region. Values were plotted using GraphPad Prism. For ChIP-PCR assays, the obtained PCR products from SM, (+), and (−) antibody were separated on 2% agarose gels and visualized on BioRad Gel Doc XR using Quantity One v4.6.9.

**Microscopy and analysis.** Overnight cultures grown in YPD were subcultured into fresh YPD at 0.1 OD$_{600}$ and grown until 0.4–0.6 OD$_{600}$. Cells were isolated, washed twice in 1× PBS, and mounted on slides. Images for Supplementary Fig. 1k (CNAG_01340) were acquired using the Airyscan mode in the Zeiss LSM 880 confocal system equipped with an Airyscan module, 63× Plan Apochromat 1.4 NA. Z stacks were obtained at an interval of 166 nm, 488/516- and 561/595-nm excitation/emission wavelengths were used as GFP and mCherry, respectively. Airyscan images were processed using Zen Black v2.3 (Zeiss) and visualized in ImageJ[78,79]. Images for Fig. 1d and Supplementary Fig. 1f, k were acquired in the Zeiss LSM 880 confocal system equipped with GaAsp photodetectors. Z stacks were obtained at an interval of 300 nm, 488 nm, and 561-nm excitation was used for GFP and

mCherry, respectively, and emission between 490–553 nm and 571–651 nm was captured. Images are represented as maximum-intensity projections.

Live-cell microscopy, images for kinetochore quantitation, and microscopy-based assays were acquired using the Zeiss Axio Observer 7, equipped with Definite Focus.2, Colibri 7 (LED light source), TempController 2000-2 (PECON), 100× Plan Apochromat 1.4 NA objective, pco.edge 4.2 sCMOS, and Andor iXon Ultra 897 electron-multiplying charge-coupled device (CCD). Zen 2.3 (blue edition) was used for image acquisition and controlling all hardware components. Filter set 92 HE with excitation 455–483 and 583–600 nm for GFP and mCherry, respectively, and corresponding emission was captured at 501–547 and 617–758 nm. To limit the time taken for an image, a complete Z stack was obtained for each channel before switching.

For live-cell microscopy, an overnight culture grown in YPD was subcultured into fresh YPD at ~0.1 OD$_{600}$ and grown for 2–3 generations until 0.4–0.8 OD$_{600}$. Cells were harvested, washed in 1× PBS, and resuspended in synthetic complete media with 3% dextrose. Cells were mounted onto an agarose pad (3% dextrose, 3% agarose in synthetic complete media) and sealed with petroleum jelly. Images were captured at time intervals of 0.5, 1, 2, or 4 min, as appropriate, with an EM gain of 300 and Z interval of 300 nm. Z-stack projection of images is represented. For live-cell quantitation of kinetochore signal, signal intensities were measured after the projection of Z stacks.

To study kinetochore interdependency, *GAL7p* conditional strains were grown overnight in YPG, subcultured at 0.2 OD$_{600}$, and grown until 0.8–1 OD$_{600}$. Cells were washed and resuspended in 1× PBS. Following which cells were inoculated into YPD (repressive, −) and YPG (permissive, +) at 0.1 OD$_{600}$. Timepoints chosen for the microscopic analysis of the repression phenotype were based on when the repressed kinetochore protein signals were no longer detectable at the kinetochore, which often coincided with an ~90% loss in cell viability (Supplementary Fig. 2g). Images were acquired after 6, 12, 15, 18, 9, and 18 h for CENP-C$^{Mif2}$, Mis12$^{Mtw1}$, Nuf2, Knl1$^{Spc105}$, Dad1, and Dad2, respectively. Z stack was obtained at an interval of 300 nm. Single Z slice representing the maximum intensity of the tagged kinetochore proteins was represented. Quantitation of kinetochore signal was performed from large budded cells (budding index ~0.55–0.90).

To estimate the population of large buds and cells with segregation defects, cells were grown until early-log phase 0.8–1 OD$_{600}$ after subculture from an overnight culture. They were imaged using the above-mentioned sCMOS camera with a Z interval of 300 nm. Cells with a budding index of >0.55 were considered as large budded cells in mitosis. Chromatin marked with a GFP/mCherry-tagged histone H4 construct was used to observe missegregation events. In total, >1000 cells were measured for each independent experiment to estimate segregation defects. To calculate the percent large budded population after 10 μg/ml TBZ treatment, >300 cells were measured for each independent experiment.

Images for the OE assay of bridgin strains are representative maximum-intensity projection images. KMN network and Dam1C components reach the peak intensity at metaphase, while bridgin exhibits maximum kinetochore intensity at anaphase. For quantitation of kinetochore signals, measurements were performed at the M phase to capture their signal at peak intensity, where possible. Components of the outer kinetochore are transiently localized to the kinetochore from G2 to M. Thus, clustered kinetochore signals in the daughter bud (budding index of ~0.55–0.8) provided us with a reliable stage to score for the kinetochore signal intensities. To maintain uniformity and ease of measurement, bridgin and inner kinetochore signal intensities were also measured at metaphase. Kinetochore signal measurement in interdependency assays and *bgi1Δ* background were measured from the in-focus Z plane exhibiting the most intense signal. Background signal measured from a region neighboring the kinetochore-measured signal in the same plane of the equal area was subtracted from the measured kinetochore intensity and normalized to the appropriate control and plot using GraphPad Prism v5.00 or v7.0a (GraphPad software). All acquired images were processed in ImageJ[78,79]. For images wherein brightness and contrast are modified, the settings were applied uniformly across the entire image.

**Budding index calculation.** Budding index of a cell is defined as the ratio obtained by

(1) The diameter of the daughter cell/diameter of the mother cell

The diameter of the daughter and mother cell was measured along the mother–daughter axis using the line tool in ImageJ[79].

**Generation of recombinant proteins.** GST, GST-Bgi1$^{FD}$ (residues 1–130), and GST-Bgi1$^{BD}$ (residues 1000–1295) were expressed from pGEX-6P-1 (GE Healthcare) in Rosetta2 (DE3) (Merck). GST and GST-Bgi1$^{BD}$ were induced for expression using 1 mM IPTG for 3 h at 37 °C. GST-FD was induced for expression overnight at 16 °C using 0.2 mM IPTG. Cells were harvested and lysed in lysis buffer (20 mM HEPES, pH 7.5, 300 mM NaCl, 1 mM EDTA, 0.5 mM TECP, and 1× complete EDTA-free protease inhibitor (Roche)) and 1× PBS with 1× complete EDTA-free protease inhibitor (Roche) for GST and GST-Bgi1$^{FD}$. GST fusion proteins were affinity purified using Glutathione sepharose 4b beads (GE Healthcare) and eluted using 20 mM glutathione. GST-Bgi1$^{FD}$ and GST-Bgi1$^{BD}$ were further purified using anion exchange chromatography. The column was

equilibrated using 20 mM Tris-HCl pH 7.5, 1 mM DTT. An elution gradient of 5–75% NaCl was achieved using 20 mM Tris-HCl, pH 7.5, 1 M NaCl, and 1 mM DTT. Relevant fractions were pooled, concentrated in Amicon-Ultra (Merck), frozen in liquid nitrogen, and stored at −80 °C.

His-Bgi1$^{FL}$ was expressed in SF9 cells. Cells were resuspended and lysed in binding buffer (20 mM Tris-HCl, pH 8.0, 500 mM NaCl, and 5 mM imidazole). His-Bgi1$^{FL}$ was affinity purified with Ni-NTA agarose (GE Healthcare), eluted with 20 mM Tris-HCl, pH 8, 500 mM NaCl, and 500 mM imidazole. Purified protein was dialyzed against buffer containing Tris-HCl, pH 7.5, 1 mM DTT, and 100 mM NaCl. Samples were concentrated using Amicon-Ultra (Merck), frozen in liquid nitrogen, and stored at −80 °C. The absence of contaminating DNA was confirmed in all recombinant protein samples.

**Viability assay**. An overnight culture was inoculated into fresh YPD medium at 0.1 $OD_{600}$ and grown to ~0.8 $OD_{600}$. The cell number was measured, followed by dilution of the cell suspension. Multiple dilutions containing 100–500 cells were subsequently plated on YPD solidified using 2% agar and grown for 2 days at 30 °C. The number of colonies formed was measured and plotted as normalized values to the wild-type control.

**Serial dilution growth analysis**. Cells were grown overnight, inoculated into fresh YPD at 0.2 $OD_{600}$, and grown until 0.8–1 $OD_{600}$. Following which cells were isolated and made up to 2 OD/ml in 1× PBS. Further dilutions were made as indicated in 1× PBS. In all, 2 µl of the cell suspension was transferred onto appropriate agar plates as mentioned and incubated for 2 days for 30 °C, DMF control and 2 µg/ml TBZ, 3 days for 30 °C + 4 µg/ml TBZ, and 37 °C and 7 days for 14 °C.

**Electrophoretic mobility shift assays**. Purified recombinant proteins of the mentioned molar ratio were incubated with 601 DNA (2.5 pM) or 1 pM of reconstituted nucleosomes in binding buffer (20 mM Tris, pH 7.5, 100 mM NaCl, 5% glycerol, and 1 mM DTT). They were incubated for 1 h at 4 °C and separated on a PAGE gel, stained with GelRed, and visualized using a UVP gel documentation system equipped with Doc-It LS Image acquisition software (UVP) v6.7.1. Further, the gels were stained with Coomassie to visualize the protein complexes and imaged using a scanner.

To generate chicken or human nucleosomes, histone H3.2 and histone H4 tetramer or CENP-A and histone H4 tetramer were mixed with histone H2A/H2B dimer with 601 DNA. KCl was added to the mixture to a final concentration of 2 M. The subsequent mixture was dialyzed against a salt gradient buffer (2 M to 200 mM KCl, 10 mM Tris-HCl, pH 7.5, 1 mM EDTA, and 10 mM DTT) overnight at 4 °C. The mixture was dialyzed to the final buffer (100 mM KCl, 10 mM Tris-HCl, pH 7.5, 1 mM EDTA, and 10 mM DTT) for 3 h at 4 °C. The pellet in the mixture was removed by centrifugation at 5000g at 4 °C. To stabilize the resultant nucleosome, the supernatant was kept in the incubator at 37 °C for 1 h. The reconstituted nucleosomes were checked by native-PAGE and detected by EtBr staining for DNA.

**Estimation of DNA methylation**. Genomic DNA was isolated from overnight cultures of wild type and bgi1Δ, using a modified glass bead protocol[76]. In brief, cells were suspended in a microfuge tube containing 500 µl of lysis buffer (50 mM Tris-HCl, pH 7.5, 20 mM EDTA, and 1% SDS) and 250 µl of glass beads. Cells were disrupted by vortexing for 5 min and centrifuged for 1 min at 15,000g. To the supernatant, 275 µl of 7 M ammonium acetate was added and incubated at 65 °C for 5 min and rapidly chilled on ice for 5 min. In all, 500 µl of chloroform was added, mixed, and centrifuged at 15,000g for 3 min. The supernatant containing DNA was precipitated with isopropanol, washed with 70% ethanol, dried, and resuspended in 50 µl of deionized water.

The isolated genomic DNA was digested separately with CpG methylation-sensitive (HhaI) or insensitive (HindIII) restriction enzymes (New England Biolabs) overnight with a no enzyme (uncut) control reaction. The digested DNA was diluted 1:50 and used for PCR amplification. Primer sets for PCR amplification of the centromeric region (5′-AGTCTCGTGTGGCTATGATT-3′ and 5′-GGATCT GCTTGACAGTGTCA-3′) and noncentromeric regions (5′-CCAACCGAAGCC CAAGACAA-3′ and 5′-TTGAAGGATGATCCGGCCGA-3′) were used. The obtained PCR products were subsequently separated by agarose gel electrophoresis using a 1% agarose gel and visualized by EtBr staining.

**Statistics and reproducibility**. Standard deviation and the mean of at least three independent experiments are mentioned in all plots unless mentioned otherwise. Statistical significance of differences was calculated as mentioned in the figure legends with one-way ANOVA with Dunnett's or Tukey's multiple comparison test, Mann–Whitney two-tailed, or Kruskal–Wallis one-way analysis followed by Dunn's multiple comparisons test. P values ≥ 0.05 were considered as non-significant (n.s.). Precise p values are mentioned within figures if significant and above p = 0.0001. All analyses were conducted using Microsoft Excel v2008 or GraphPad Prism version Windows v5.00 or v7.0a. Micrographs and

immunoblotting images shown in figures are representative of at least two independent experiments with similar results.

**Reporting summary**. Further information on research design is available in the Nature Research Reporting Summary linked to this article.

## Data availability

The source data underlying Fig. 1c, Figs. 2a, c–f, 3b, c, e, Fig. 4a, b, f, Fig. 5d, e, Fig. 6a, b, Fig. 7c, d, g, h, Supplementary Fig. 1a–c, Supplementary Fig. 2a, b, d, e, g, Supplementary Fig. 3a–j, Supplementary Fig. 4a, c, Supplementary Fig. 5c, e, f, Supplementary Fig. 6a–d, f, h, and Supplementary Fig. 7c–i are provided in the Source Data file. All MS proteomics data that are presented in this study have been deposited to the ProteomeXchange Consortium via jPOST with the dataset identifiers "PXD021072" and "JPST000947", respectively. Publicly available databases used in the study include C. neoformans H99 proteome database (https://www.uniprot.org/proteomes/UP000010091) and Uniprot proteome database (https://www.uniprot.org/proteomes/).

Yeast strains and other data that support the findings of this study are available upon reasonable request from the corresponding authors. Source data are provided with this paper.

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

## Acknowledgements

The authors thank the members of the Sanyal and Fukagawa laboratories for their inputs and discussions. We thank Daniel Gerlich (Institute of Molecular Biotechnology, Vienna, Austria) for generously providing us the Ki67 plasmids. The authors are grateful to Masatoshi Hara, Mariko Ariyoshi, Reito Watanabe, and Fumiaki Makino from the Fukagawa laboratory for their technical assistance. We also thank Akira Shinohara and his group members for allowing us to use their facilities. We acknowledge B. Suma at the confocal imaging facility, JNCASR, for assistance in imaging. A joint grant from the Department of Science and Technology (DST), India, and Japan Society for the Propagation of Science (JSPS), Japan aided in the travel of S.S. between the Sanyal and Fukagawa laboratories (DST/INT/JSPS/P-240/2017). Research in the Sanyal lab was funded by the grants from DST, Govt. of India (GOI) to KS (CRG/2019/005549), Department of Biotechnology (DBT), GOI at JNCASR (BT/INF/22/SP/27679/2018), JC Bose National Fellowship from Science and Educational Research Board, GOI, (JCB/2020/000021), and intramural funding from JNCASR to KS. KS was a Tata Innovation Fellow DBT, GOI (BT/HRD/35/01/03/2017). S.S. thanks Council for Scientific and Industrial Research (09/733(0192)/2014-EMR-I) and JNCASR for his fellowship. Research in the Fukagawa laboratory was supported by JSPS KAKENHI Grant nos. 17H06167, 16H06279, and 15H05972 and Osaka University International Joint Research Promotion Program type A.

## Author contributions

S.S. and K.S. conceived the project. S.S., K.S., T.H., and T.F. designed the experiments. R. N. performed the mass spectrometry and S.S. performed the rest of the experiments. T.H. assisted with affinity purification for mass spectrometry. S.S. and K.S. wrote the paper. S.S., K.S., T.F., and T.H. revised the paper. All authors approved the final version. K.S. gathered funding and supervised the entire project.

## Competing interests

The authors declare no competing interests.
