## [Peer Review File · Nature Communications]

Reviewers' comments:

Reviewer #1 (Remarks to the Author):

High-fidelity chromosome segregation is a fundamental process for all the organisms, and the evolution of kinetochore components involved in the process in eukaryotes has been attracting attention in recent years. Intriguingly, in contrast to the conservation of the KMN(KNL1, Mis12, and Ndc80 complexes) network components at the outer kinetochore in a wide range of species, the CCAN(constitutive centromere associated network) components at the inner kinetochore are often missing in various species. Different lineages sometimes use distinct components for a similar role, such as the Ska complex in vertebrates and the Dam1 complex in yeasts. However, a little is known whether there are additional kinetochore components in other eukaryotes that are absent in the popular model organisms.

In this manuscript, the authors reported a novel kinetochore protein, named bridgin, which they found in a human fungal pathogen, *Cryptococcus neoformans*. They identified bridgin as a common interactor of known kinetochore proteins CENP-C(Mif2), Dsn1, and Spc25. Bridgin was found to localize to kinetochores from G2 to telophase, peaking at anaphase, and the localization is dependent on Mis12(Mtw1) and partly on Sos7, but not on Dad2. On the other hand, kinetochore localization of Nuf2, KNL1(Spc105), Mis12, and CENP-A(Cse4) was not affected by bridgin deletion. Bridgin-deleted cells were viable, but showed reduced growth and an increase in the rate of chromosome missegregation, which is supposedly due to defective kinetochore-microtubule interactions. Bridgin is a 1295aa protein comprised of the large unstructured domain (USD) flanked by the N-terminal FHA(fork-head associated) domain (FD) and the C-terminal basic domain (BD). Observation of deletion mutants localization of bridgin showed that both FD and USD are responsible for the kinetochore localization, while BD is dispensable. In contrast, complementation assays showed that all the domains are required for cell growth and faithful chromosome segregation. BD binds to DNA and nucleosomes in in vitro assays, and a bridgin overexpression experiment suggested that BD binds non-specifically to chromatin, not restricted to centromeric chromatin. Interestingly, replacing BD of bridgin with the similarly basic region of human Ki67 fully complemented cell growth and chromosome missegregation. From these data, the authors concluded that bridgin ensures mitotic fidelity by linking the outer kinetochore to centromeric chromatin. Intriguingly, bridgin homologues are identified throughout basidiomycete sub-phylum, and also outside of fungi, suggesting a more ancient origin of the protein. Furthermore, Ki67 has a bridgin-like structure and may be a metazoan counterpart of bridgin.

Overall, this study provides valuable insights into kinetochore evolution and a general framework for kinetochore structure, and stresses the importance of studying organisms other than standard model organisms. Bridgin is a novel linker protein having a unique feature that it is recruited by outer kinetochore components and probably localizes to centromeric chromatin through non-specific chromatin binding. It is exciting that bridgin is conserved widely in eukaryotes and might be repurposed as Ki67 in vertebrates. The study is comprehensive and convincing as an initial report on this novel kinetochore component, which is of interest to general readers. The following are suggestions that would help further understanding of the readers.

Major points:

1. Dependency of kinetochore binding of bridgin on outer kinetochore components is a crucial

finding in the study. Although detailed biochemical analyses to define the interaction of FD and USD with particular outer kinetochore components are beyond the focus of the study, more interdependency analyses at a cellular level would be informative, especially because only a partial reduction of bridgin kinetochore localization was seen in *Sos7*-deleted cells. As the reviewer could not find sufficient information that *Sos7* is a genuine component of the KNL1 complex, please provide the reference or characterization regarding the point, and show the dependency of bridgin kinetochore localization on KNL1 itself.

2. Regarding the kinetochore localization of bridgin in mitosis, is it expressed throughout the cell cycle and relocated to centromeric chromatin upon outer kinetochore assembly? The reviewer recommends an immunoblot analysis of bridgin during the cell cycle, which is possible by detecting the GFP-tag.

3. Full complementation of bridgin function by replacing BD with the basic region of human Ki67 is striking. However, if Ki67 is a vertebrate counterpart of bridgin, conserved features other than enrichment of positively-charged residues might be involved. It would be informative to show sequence similarity between BD of bridgin and corresponding human Ki67 region whether there are conserved motifs or residues.

Minor points:

1. page 8, line 12: The description “unattached chromosomes” should be reconsidered, because it cannot be judged without microtubule visualization.

2. page 10, line 25: The description “Microtubule-like signal... Hence, we ruled out the possibility...” should be reconsidered, as they are not sufficient to exclude the possibility that bridgin binds to microtubules.

3. Schematic domain architecture of bridgin in different basidiomycete species, such as shown in Figure 1b, would be helpful to have an idea on their structural conservation.

Reviewer #2 (Remarks to the Author):

The kinetochore is a multi-protein complex that drives chromosome segregation in eukaryotes. Extensive studies in select model organisms have identified a number of kinetochore components, and bioinformatic analyses have shown that some kinetochore proteins are widely conserved among eukaryotes. However, there is only a limited number of experimental studies on kinetochores in non-traditional model eukaryotes and it therefore remains unclear whether there are lineage-specific kinetochore components because such proteins cannot be identified by means of bioinformatics using known kinetochore proteins as query. To identify kinetochore components in the basidiomycete *Cryptococcus neoformans* (an important human pathogen that apparently lacks most of CCAN components except Mif2), the authors performed immunoprecipitation of three kinetochore proteins (Mif2, Spc25, and Dsn1) and identified co-purifying proteins by mass spectrometry. This led to the identification of a previously unknown kinetochore protein, which they

named bridgin (Bgi1). Bgi1 co-localizes with kinetochore proteins from G2 to anaphase, with its signal peaking during anaphase. Kinetochore localization of Bgi1 depends on Mtw1 and Sos7, but not Dad2. The bgi1 deletion mutant is viable but has an increased mis-segregation rate and delay in mitotic progression. Using truncation proteins, the authors showed that Bgi1 has at least two domains (FHA domain and unstructured domain) that can promote kinetochore localization, while the C-terminal basic domain can bind DNA non-specifically in vitro. All domains are indispensable for the Bgi1 function. Importantly, the DNA-binding domain of Ki-67 can functionally replace that of Bgi1.

Overall, this is a very interesting and important study identifying and characterizing a lineage-specific kinetochore protein in *C. neoformans*. This work should encourage more researchers to study non-traditional model eukaryotes to better understand the function and evolution of kinetochores. Experiments in this manuscript were carried out to a high standard. I strongly support publication of this manuscript in Nature Communications if the authors can address the following comments.

Major comments

1. While this manuscript reports a very nice characterization of a newly identified kinetochore protein bridgin, I do not find convincing evidence in this manuscript that this protein acts as a “linker”. The term “linker kinetochore protein” could have different meanings to different people, so it should be clearly defined by the authors. In this manuscript, the authors apparently use it as an inner kinetochore protein that directly binds and recruits KMN proteins (i.e. CENP-C, CENP-T, Ame1). Although they show that Bgi1 has DNA-binding activities in vitro and localizes at kinetochores in the Mtw1/Sos7-dependent manner in vivo, the bgi1 deletion mutant did not affect the localization of KMN proteins. Without providing direct protein-protein interactions between Bgi1 and KMN proteins or recruitment of KMN by Bgi1, for example, by the tethering assay previously used by the Fukagawa lab and others, I do not think it is proper to call Bgi1 as a “linker” protein (of their definition). The title/abstract/manuscript should be changed accordingly. Alternatively, they may change the definition of linker proteins.

Minor comments

1. As the authors' analyses imply, it seems that bridgin shares common ancestry with Ki-67, based on the presence of an N-terminal FHA domain, PP1-binding motif, and C-terminal basic domain in both proteins. It would be important to emphasize that Ki-67 is not a kinetochore protein in humans and that bridgin and Ki-67 have distinct functions (i.e. neofunctionalization), which is an interesting discovery.
2. Figure 1c: why are there arrows of two (not one or three) different colors?
3. Figure 5b is confusing. The text says “No detectable association of histone H4 was obtained with ... or Bgi1 BDA (Fig. 5c: immunoblots)” but Figure 5b (mass spec data) shows that H4 is present in Bgi1 BDA. Instead of showing the relative abundance of a protein in a given sample, they should normalize the protein abundance between samples or provide a table like Figure 1d.
4. Although Bgi1 BD clearly binds DNA in vitro, it is less convincing that Bgi1 BD binds DNA/chromatin in vivo. Have the authors performed IP/MS for Bgi1 BD to see if it co-purifies with histones and/or chromatin proteins?
5. Supplementary figure 4 g-j: Have the authors quantified the signal of these kinetochore proteins in anaphase cells, when Bgi1 signal is the highest?
6. page 7 line 28 etc: Bridgin is “essential” for accurate kinetochore-microtubule interaction. Given

that the *bgi1* deletion mutant is viable, I feel it is more proper to use the word “important”.

7. I found a number of sentences in Discussion with grammar problems or typos (e.g. page 12 lines 14, 22). I suggest the authors carefully go through the whole manuscript.

Reviewed by Bungo Akiyoshi

Reviewer #3 (Remarks to the Author):

Sridhar et al. identify a novel protein, *Bgi1*, in *C. neoformans* via computational and proteomic approaches and characterize its involvement in kinetochore assembly and microtubule interaction using tagged strains, inducible systems to assess epistatic and assembly-dependent interactions, and fluorescent live cell imaging to assess localization during the course of the cell cycle. Overall, the findings add to our understanding of *C. neoformans* cell biology, and they place this work in the larger context of fungal kinetochore structure. While the experimental design appears appropriate, and interpretations of the data are consistent with the underlying assumptions, many of the conclusions about this novel protein depend on conserved function of proteins that have not yet been individually characterized in this system (such as *Mad2*). Given the large-scale changes between *C. neoformans* and more thoroughly characterized systems (including apparent loss of CENP-T and CENP-Q complexes which the authors identify), reliance on these assumptions may undermine the overall model. Despite this weakness, the authors have substantially improved on existing models of kinetochore assembly in *C. neoformans*.

General comments:

There are numerous grammatical errors throughout, including in the title.

All abbreviations must be defined at the first time of use and in the relevant figure legends.

Figure order should reflect when they are discussed in the text (see p 6 line 27 for example)

Genotypes of represented strains must be provided in the main text, not just the supplementary tables. Readers should not have to dig through supplementary files just to understand main figures.

In many places, the authors fail to describe experimental approaches and rationale in sufficient detail, instead leaping straight to interpretation. This is most problematic in the section on recruitment and assembly, but plagues other areas as well. There is also crucial information required for understanding the major findings buried in supplemental figures that lack experimental description.

According to the methods, t-tests were used for all statistics, but this is not appropriate for multiple comparisons such as those presented in Figure 3C and elsewhere. This makes it impossible to assess much of the data with confidence. Data should be assessed for normality and the appropriate test applied.

In some cases there is a lack of replication that must be addressed (Supplemental fig 5, for example) or a lack of experimental detail for concepts described in the main text that are presented as

supplemental figures. Low cell numbers (<100 per biological replicate) and a lack of justification for why particular cells were selected for analysis further raise potential issues with data analysis that must be addressed.

Regarding interdependency analyses, low cell numbers for strains using the GAL inducible system, coupled with arbitrary imaging times and the lack of evidence of glucose-mediated repression following galactose induction in *Cryptococcus grubii* invalidate findings using this reporter system. The authors must demonstrate that protein expression is in fact repressed, for example through biological consequence of repression (mis-segregation analysis, altered growth, sensitivity phenotypes). They should include imaging times for the indicated strains in the figure legends, as well as rational for those times in the main text. Note that Wickes et al., who developed this reporter, did so using *C. neoformans* var. *neoformans* and did not demonstrate that it is repressible following induction, only that it is not induced in the absence of galactose. Others using this system in *Cn. grubbii* have observed leaky expression in glucose and have observed differences in induction (Baker and Lodge 2012).

Reviewer #4 (Remarks to the Author):

Review

Sridhar and colleagues show that several fungal lineages lack known kinetochore linker complexes and use proteomics to identify possible novel linker proteins in the basidiomycete *C. neoformans*. They find several candidates that were present in all pulldown experiments of three known kinetochore components, and zoom in on one based on its co-localization with a kinetochore marker. This candidate they name bridgin and the remainder of the manuscript details its functional and biochemical characterization. In short:

- its localization to kinetochores depends on the ndc80 complex
- its deletion from the *Cryptococcus* genome causes chromosome segregation errors
- both of its definable features (FHA domain, basic domain) are required for function
- Its basic domain binds aspecifically to DNA
- function of the basic domain depends not on its sequence but on its charge.

This is an interesting study that combines observation from comparative genomics (loss of CCAN complexes) with experimental investigations to explain those losses (identification and

characterization of a new kinetochore protein). The data convincingly show bridgin is a kinetochore protein and that it is important for accurate chromosome segregation. The manuscript could do however with more insight into bridgin's molecular function.

My main comment is that bridgin is presented as a bridge between chromatin and the outer kinetochore, but its function is actually quite poorly examined. For example, loss of known linker complexes in budding yeast or animal cells prevent the outer kinetochore from assembling properly. In bridgin's case, that is not examined. In fact, bridgin depends on the Mis12- and Knl1 complexes (not sure why they conclude that it depends on the Ndc80 complex (fig 2g): that is not in fact tested). If the KMN network does not depend on bridgin, it can hardly be a linker complex. In fact, it would mean that *Cryptococcus*' most essential linker 'activity' is still mysterious. Then there is the observation that bridgin binds DNA and chromatin aspecifically. It is not easy to imagine how that is a useful characteristic of a linker protein. Does it bind centromeric sequences or chromatin with higher affinity than non-centromeric ones? Does it need DNA binding? The fact that deleting BD compromises function and can be replaced with the BD of Ki67 is in my view not sufficient evidence that it is the DNA-binding capacity of BD that is needed for bridgin function.

In addition: most elements of the model in 6g are not shown: does the FD bind outer kinetochore proteins? Does the BD bind centromeric DNA? What can be a molecular explanation for why BD does not bind DNA unless FD binds the kinetochore? Bridgin is proposed to promote stable kinetochore-microtubule interactions, but that is inferred mostly from the observations that bridgin mutants delay in mitosis in a mad2-dependent manner and that they show missegregations. These can be explained by many other possible defects.

In short, I feel the finding is potentially interesting but quite preliminary at this stage.

Point-by-point response

Sridhar *et al.*

Nature Communications (MS# NCOMMS-19-32938A)

We are grateful to all reviewers and the editor and their positive remarks, constructive criticisms, and suggestions on our study. Kindly find below our point-by-point response to each comment.

Reviewer 1

High-fidelity chromosome segregation is a fundamental process for all the organisms, and the evolution of kinetochore components involved in the process in eukaryotes has been attracting attention in recent years. Intriguingly, in contrast to the conservation of the KMN (KNL1, Mis12, and Ndc80 complexes) network components at the outer kinetochore in a wide range of species, the CCAN (constitutive centromere associated network) components at the inner kinetochore are often missing in various species. Different lineages sometimes use distinct components for a similar role, such as the Ska complex in vertebrates and the Dam1 complex in yeasts. However, a little is known whether there are additional kinetochore components in other eukaryotes that are absent in the popular model organisms.

In this manuscript, the authors reported a novel kinetochore protein, named bridgin, which they found in a human fungal pathogen, *Cryptococcus neoformans*. They identified bridgin as a common interactor of known kinetochore proteins CENP-C(Mif2), Dsn1, and Spc25. Bridgin was found to localize to kinetochores from G2 to telophase, peaking at anaphase, and the localization is dependent on Mis12(Mtw1) and partly on Sos7, but not on Dad2. On the other hand, kinetochore localization of Nuf2, KNL1(Spc105), Mis12, and CENP-A(Cse4) was not affected by bridgin deletion. Bridgin-deleted cells were viable but showed reduced growth and an increase in the rate of chromosome missegregation, which is supposedly due to defective kinetochore-microtubule interactions. Bridgin is a 1295aa protein comprised of the large unstructured domain (USD) flanked by the N-terminal FHA (fork-head associated) domain (FD) and the C-terminal basic domain (BD). Observation of deletion mutants localization of bridgin showed that both FD and USD are responsible for the kinetochore localization, while BD is dispensable. In contrast, complementation assays showed that all the domains are required for cell growth and faithful chromosome segregation. BD binds to DNA and nucleosomes in *in vitro* assays, and a bridgin overexpression experiment suggested that BD binds non-specifically to chromatin, not restricted to centromeric chromatin. Interestingly, replacing BD of bridgin with the similarly basic region of human Ki67 fully complemented cell growth and chromosome missegregation. From these data, the authors concluded that bridgin ensures mitotic fidelity by linking the outer kinetochore to centromeric chromatin. Intriguingly, bridgin homologues are identified throughout basidiomycete sub-phylum, and also outside of fungi, suggesting a more ancient origin of the protein. Furthermore, Ki67 has a bridgin-like structure and may be a metazoan counterpart of bridgin.

Overall, this study provides valuable insights into kinetochore evolution and a general framework for kinetochore structure, and stresses the importance of studying organisms other than standard model organisms. Bridgin is a novel linker protein having a unique feature that it is recruited by outer kinetochore components and probably localizes to centromeric chromatin through non-specific chromatin binding. It is exciting that bridgin is

conserved widely in eukaryotes and might be repurposed as Ki67 in vertebrates. The study is comprehensive and convincing as an initial report on this novel kinetochore component, which is of interest to general readers.

Au: We thank the reviewer for encouraging general comments on our work and the words of appreciation.

The following are suggestions that would help further understanding of the readers.

Major comments

R1-1: Dependency of kinetochore binding of bridgin on outer kinetochore components is a crucial finding in the study. Although detailed biochemical analyses to define the interaction of FD and USD with particular outer kinetochore components are beyond the focus of the study, more interdependency analyses at a cellular level would be informative, especially because only a partial reduction of bridgin kinetochore localization was seen in *Sos7*-deleted cells. As the reviewer could not find sufficient information that *Sos7* is a genuine component of the KNL1 complex, please provide the reference or characterization regarding the point, and show the dependency of bridgin kinetochore localization on KNL1 itself.

Au: A) We provided the following additional information on *Sos7* as a component of the Knl1 complex:

i) The confidence values and sequence conservation of *C. neoformans* Cn*Sos7* with *Schizosaccharomyces pombe* Sp*Sos7*, as well as other identified homologs are shown (new Supplementary Fig. 1d and e). Sp*Sos7* was initially identified as a conserved protein that exhibits genetic and tight physical interactions with Spc105^{Knl1}¹. Further, analysis by the Kops group² suggested that the Knl1^{Spc105} interacting proteins in eukaryotes, *Sos7*, Zwint, and Kre28 belong to the same orthologous group.

ii) In the previous version of this manuscript, we reported the G2/M specific kinetochore localization of *Sos7* (Fig. 1e and Supplementary Fig. 1f). In the revised manuscript, we added new data showing that *Sos7* is dispensable for viability but plays a critical role in accurate chromosome segregation (new Supplementary Fig. 1g and h).

Based on strong sequence homology to other *Sos7* homologs in fungi, and its cell-cycle dependent kinetochore specific localization patterns similar to those observed for KMN network proteins, we conclude that the protein coded by the ORF CNAG_03715 is the *C. neoformans* *Sos7* homolog.

B) As suggested by reviewer 1, the dependency of bridgin on Knl1^{Spc105} was tested in the revised manuscript (new Fig. 2e). It was observed that bridgin was found to be partially reliant (~30%) on Knl1^{Spc105} for its kinetochore localization. Due to technical limitations, we could not reliably determine the dependence of bridgin simultaneously on Knl1^{Spc105} and *Sos7*. However, bridgin is more reliant on *Sos7* (Fig. 2f) than Knl1 (new Fig. 2e) for its kinetochore localization. The influence of *Sos7* on other KMN network components needs to be analyzed in the future.

R1-2: Regarding the kinetochore localization of bridgin in mitosis, is it expressed throughout the cell cycle and relocated to centromeric chromatin upon outer kinetochore assembly? The reviewer recommends an immunoblot analysis of bridgin during the cell cycle, which is

possible by detecting the GFP-tag.

Au: This is an interesting possibility raised by the reviewer. Cell cycle synchronization followed by the release of *C. neoformans* cells is technically challenging and not yet established. Instead of synchronizing cells at a specific stage of the cell cycle, we arrested them in early S phase by hydroxyurea (HU) treatment, where no bridgin localization could be observed (Supplementary Fig. 1k and 2a). When we arrested cells at M phase by thiabendazole (TBZ), centromere-specific localization of bridgin was observed (Fig. 2a and Supplementary Fig. 1k). Additionally, levels of bridgin were assessed in cells collected at the log and stationary phase of growth. We observed relatively consistent levels of bridgin throughout the measured stages (new Supplementary Fig. 2b) that suggested that the localization of bridgin at the kinetochore is unlikely to be regulated by the cell cycle stage-specific levels of protein abundance in the cell.

R1-3: Full complementation of bridgin function by replacing BD with the basic region of human Ki67 is striking. However, if Ki67 is a vertebrate counterpart of bridgin, conserved features other than enrichment of positively-charged residues might be involved. It would be informative to show sequence similarity between BD of bridgin and corresponding human Ki67 region whether there are conserved motifs or residues.

Au: We compared the sequence of BD of bridgin and the corresponding Ki67 region generated and presented in Supplementary Fig. 7a of the revised manuscript. Between the basic domain of bridgin and Ki67 of humans and other vertebrate species, the sequence conservation is minimal and generally restricted to a few basic residues (highlighted in red). Further, our extensive bioinformatic analysis of aligned basic domain sequences of basidiomycetes or metazoa failed to identify the homolog in the other, suggesting weak sequence conservation between them. We also described this point in the revised text.

Minor comments

R1-4: page 8, line 12: The description “unattached chromosomes” should be reconsidered, because it cannot be judged without microtubule visualization.

Au: We agree with the reviewer’s suggestion, and the statement was removed from the text in the revised manuscript.

R1-5: page 10, line 25: The description “Microtubule-like signal.... Hence, we ruled out the possibility....” should be reconsidered, as they are not sufficient to exclude the possibility that bridgin binds to microtubules.

Au: We agree with the reviewer’s suggestion, and the statement was removed from the text in the revised manuscript.

R1-6: Schematic domain architecture of bridgin in different basidiomycete species, such as shown in Figure 1b, would be helpful to have an idea on their structural conservation.

Au: We thank the reviewer for raising this point. We have included the domain architecture of identified bridgin homologs across basidiomycetes as Fig. 7b in the revised manuscript.

Reviewer 2

The kinetochore is a multi-protein complex that drives chromosome segregation in eukaryotes. Extensive studies in select model organisms have identified a number of kinetochore components, and bioinformatic analyses have shown that some kinetochore proteins are widely conserved among eukaryotes. However, there is only a limited number of experimental studies on kinetochores in non-traditional model eukaryotes and it therefore remains unclear whether there are lineage-specific kinetochore components because such proteins cannot be identified by means of bioinformatics using known kinetochore proteins as query. To identify kinetochore components in the basidiomycete *Cryptococcus neoformans* (an important human pathogen that apparently lacks most of CCAN components except Mif2), the authors performed immunoprecipitation of three kinetochore proteins (Mif2, Spc25, and Dsn1) and identified co-purifying proteins by mass spectrometry. This led to the identification of a previously unknown kinetochore protein, which they named bridgin (Bgi1). Bgi1 co-localizes with kinetochore proteins from G2 to anaphase, with its signal peaking during anaphase. Kinetochore localization of Bgi1 depends on Mtw1 and Sos7, but not Dad2. The bgi1 deletion mutant is viable but has an increased mis-segregation rate and delay in mitotic progression. Using truncation proteins, the authors showed that Bgi1 has at least two domains (FHA domain and unstructured domain) that can promote kinetochore localization, while the C-terminal basic domain can bind DNA non-specifically in vitro. All domains are indispensable for the Bgi1 function. Importantly, the DNA-binding domain of Ki-67 can functionally replace that of Bgi1.

Overall, this is a very interesting and important study identifying and characterizing a lineage-specific kinetochore protein in *C. neoformans*. This work should encourage more researchers to study non-traditional model eukaryotes to better understand the function and evolution of kinetochores. Experiments in this manuscript were carried out to a high standard. I strongly support publication of this manuscript in Nature Communications if the authors can address the following comments.

Au: We are thankful to Dr. Akiyoshi for his encouraging and highly positive remarks on the choice of problems, the quality of experimental data, and the significance of the study presented in this manuscript.

Major comments

R2-1: While this manuscript reports a very nice characterization of a newly identified kinetochore protein bridgin, I do not find convincing evidence in this manuscript that this protein acts as a “linker”. The term “linker kinetochore protein” could have different meanings to different people, so it should be clearly defined by the authors. In this manuscript, the authors apparently use it as an inner kinetochore protein that directly binds and recruits KMN proteins (i.e. CENP-C, CENP-T, Ame1). Although they show that Bgi1 has DNA-binding activities in vitro and localizes at kinetochores in the Mtw1/Sos7-dependent manner in vivo, the bgi1 deletion mutant did not affect the localization of KMN proteins. Without providing direct protein-protein interactions between Bgi1 and KMN proteins or recruitment of KMN by Bgi1, for example, by the tethering assay previously used by the Fukagawa lab and others, I do not think it is proper to call Bgi1 as a “linker” protein (of their definition). The title/abstract/manuscript should be changed accordingly. Alternatively, they may change the definition of linker proteins.

Au: Dr. Akiyoshi raises an important and relevant point. We agree with him and suitably modified the text to not to associate the term “kinetochore linker” with bridgin. Indeed, our

findings provide evidence that bridgin interacts with centromeric chromatin following its recruitment by the platform provided by the outer kinetochore KMN network. These results portray the bridgin family of proteins to employ a divergent strategy to connect outer kinetochore and centromeric chromatin, unlike defined kinetochore linker proteins such as CENP-C or CENP-T. Although our new data suggest that loss of bridgin can significantly reduce KMN network levels, additional validation, as suggested by the reviewer, is required to know how KMN protein levels are altered by bridgin (new Supplementary Fig. 7e-i). These findings could shed further light on bridgin's function at the kinetochore in the future.

Minor comments

R2-2: As the authors' analyses imply, it seems that bridgin shares common ancestry with Ki-67, based on the presence of an N-terminal FHA domain, PP1-binding motif, and C-terminal basic domain in both proteins. It would be important to emphasize that Ki-67 is not a kinetochore protein in humans and that bridgin and Ki-67 have distinct functions (i.e. neofunctionalization), which is an interesting discovery.

Au: We appreciate Dr. Akiyoshi in identifying this interesting aspect of our findings and for his suggestion. We have included the suggestions on page 12 line 23 and page 16 lines 4-7 in the revised manuscript.

R2-3: Figure 1c: why are there arrows of two (not one or three) different colors?

Au: The color codes have been mentioned in the figure legends. Blue arrows point to the CCAN bait protein CENP-C^{Mif2} while the green arrows point to the KMN network bait proteins of Dsn1 and Spc25.

R2-4: Figure 5b is confusing. The text says "No detectable association of histone H4 was obtained with ... or Bgi1 BDA (Fig. 5c: immunoblots)" but Figure 5b (mass spec data) shows that H4 is present in Bgi1 BDA. Instead of showing the relative abundance of a protein in a given sample, they should normalize the protein abundance between samples or provide a table like Figure 1d.

Au: As suggested, we prepared new tables presenting coverage % (C,%) and total spectrum counts (TSC) for each of the hits similar to Fig. 1d as new Fig. 5b and c in the revised manuscript. Data regarding tables presented in the previous Fig. 5b and c can still be found in Supplementary Table 4. While it is likely that specific histones were picked up in the Bgi1^{BDA} IP-MS and further in the co-IP experiments, we consistently observe greatly reduced levels of histone proteins in the Bgi1^{BDA} compared to Bgi1^{FL} IP's (new Fig. 6g and Supplementary Fig. 6b). We clearly described these points in the revised manuscript.

R2-5: Although Bgi1 BD clearly binds DNA *in vitro*, it is less convincing that Bgi1 BD binds DNA/chromatin *in vivo*. Have the authors performed IP/MS for Bgi1 BD to see if it co-purifies with histones and/or chromatin proteins?

Au: To support our claims that the BD can bind to chromatin *in vivo* in this revised manuscript, we performed a simultaneous FLAG co-IP and native-ChIP from MNase treated lysates of control and test strains, including Bgi1^{BD} (new Fig. 6f-h). As shown in the new Fig. 6g and h, our data show that similar to the CCAN component, CENP-C^{Mif2}, Bgi1^{FL} can effectively co-precipitate with histone H4 and ChIP at centromeres. However, histone H4 recovery was greatly reduced in IP for Bgi1^{BDA} similar to the KMN network component Spc25. Bgi1^{BD} IP could recover nucleosome components, but could not enrich centromere

DNA. Further, the addition of the Ki67^{BD} to the Bgi1^{BDΔ} construct was able to rescue the Bgi1-deficient phenotype (Fig. 6d and e) and restored its interaction with nucleosomes and centromeric enrichment (new Fig. 6g and h). Taken together these new results strengthen our findings that the BD can interact with chromatin *in vivo* while requiring the kinetochore recruitment of bridgin to assist in its specific association with centromeric chromatin.

R2-6: Supplementary figure 4 g-j: Have the authors quantified the signal of these kinetochore proteins in anaphase cells, when Bgi1 signal is the highest?

Au: As suggested by the reviewer, we measured the levels of CENP-A^{Cse4}, CENP-C^{Mif2}, Mis12^{Mtw1}, Nuf2, and Knl1^{Spc105} at anaphase, in addition to metaphase and described the results in the revised manuscript (new Supplementary Fig. 7e-i). In summary, our revised data suggest that specific KMN network proteins were altered in metaphase and anaphase, while no significant difference in the levels of inner kinetochore proteins could be detected. The observed differences were marginal but significant, and require further validation to strongly attribute biological significance to these findings.

R2-7: page 7 line 28 etc: Bridgin is “essential” for accurate kinetochore-microtubule interaction. Given that the bgi1 deletion mutant is viable, I feel it is more proper to use the word “important”.

Au: Per the suggestion from the reviewer, “essential” was modified to “important” on page 9 line 13 in the revised manuscript to refer to the role of bridgin in mediating accurate kinetochore-microtubule interactions.

R2-8: I found a number of sentences in Discussion with grammar problems or typos (e.g. page 12 lines 14, 22). I suggest the authors carefully go through the whole manuscript.

Au: Thank you for pointing it out. We have worked towards addressing the grammatical errors in the revised manuscript.

Reviewed by Bungo Akiyoshi

Reviewer 3

Sridhar et al. identify a novel protein, Bgi1, in *C. neoformans* via computational and proteomic approaches and characterize its involvement in kinetochore assembly and microtubule interaction using tagged strains, inducible systems to assess epistatic and assembly-dependent interactions, and fluorescent live cell imaging to assess localization during the course of the cell cycle. Overall, the findings add to our understanding of *C. neoformans* cell biology, and they place this work in the larger context of fungal kinetochore structure.

R3-1: While the experimental design appears appropriate, and interpretations of the data are consistent with the underlying assumptions, many of the conclusions about this novel protein depend on conserved function of proteins that have not yet been individually characterized in this system (such as Mad2). Given the large-scale changes between *C. neoformans* and more thoroughly characterized systems (including apparent loss of CENP-T and CENP-Q complexes which the authors identify), reliance on these assumptions may undermine the overall model. Despite this weakness, the authors have substantially improved on existing models of kinetochore assembly in *C. neoformans*.

Au: We thank the reviewer for identifying the merits of this study and agree with the reviewer that prior to this manuscript Mad2 and Sos7 have not been described in *C. neoformans*. However, we would like to state the following facts:

A) Most tagged kinetochore proteins and other tagged molecular markers used in this study were previously described, including the regulatable promoter *GAL7p*^{3,4}.

B) Mad2 has not been functionally characterized in *C. neoformans* but was suggested to be a highly conserved protein identified in previously bioinformatic studies^{2,5,6}. In the revised manuscript, we have experimentally described the highly conserved role of Mad2 in mediating a robust spindle assembly checkpoint (SAC) activity. We, in revised Supplementary Figure 5d and e, described the lack of spindle assembly checkpoint (SAC) activity in the absence of Mad2. As a consequence, *mad2Δ* cells are sensitive to TBZ (Fig. 3g). Additionally, in revised Supplementary 4d, we described the lack of cell cycle delay to correct for segregation defects by live-cell microscopy (Fig. 3c and Supplementary Fig. 4d). With the previous bioinformatic predictions and our described experimental data regarding the role of Mad2, we believe that there is strong evidence to conclude the role of Mad2 as a bonafide SAC protein in *C. neoformans*.

C) Characterization of Sos7 was also a concern shared by reviewer 1. We have addressed this concern with additional experiments, as mentioned above in our response to R1-1.

General comments

R3-2: There are numerous grammatical errors throughout, including in the title. All abbreviations must be defined at the first time of use and in the relevant figure legends. Figure order should reflect when they are discussed in the text (see p 6 line 27 for example)

Au: We thank the reviewer for pointing out these issues. We have worked towards addressing them in the revised manuscript as much as possible.

R3-3: Genotypes of represented strains must be provided in the main text, not just the supplementary tables. Readers should not have to dig through supplementary files just to understand main figures.

Au: As suggested, the genotypes have been incorporated into the figure legends as required throughout the revised manuscript.

R3-4: In many places, the authors fail to describe experimental approaches and rationale in sufficient detail, instead leaping straight to interpretation. This is most problematic in the section on recruitment and assembly, but plagues other areas as well. There is also crucial information required for understanding the major findings buried in supplemental figures that lack experimental description.

Au: As suggested by the reviewer, we have incorporated into the revised manuscript the rationale and detailed experiments design. We further explain supplementary data in more detail, as part of the main text in the revised manuscript. For example, the experimental details and rationale for the recruitment and assembly section can found on page 7 lines 5-13.

R3-5: According to the methods, t-tests were used for all statistics, but this is not appropriate for multiple comparisons such as those presented in Figure 3C and elsewhere. This makes it

impossible to assess much of the data with confidence. Data should be assessed for normality and the appropriate test applied.

Au: We thank the reviewer for raising this issue. In the revised manuscript, we used appropriate statistical tests to evaluate our experimental data. For example, although we used *t*-test in previous Fig. 3C, we appropriately corrected and used Kruskal-Wallis one-way analysis followed by Dunn's multiple comparison test to calculate the statistical significance of differences (the *p*-values show the difference compared to their respective controls, wild-type or *mad2Δ*) in the revised manuscript. Similar corrections were carried out across the manuscript.

R3-6: In some cases there is a lack of replication that must be addressed (Supplemental fig 5, for example) or a lack of experimental detail for concepts described in the main text that are presented as supplemental figures.

Au: 1) We incorporated detailed experimental protocols wherever necessary.
 2) The issue of a lack of replication (for example, Supplementary Fig. 6 in the current format) has been addressed. Replication of the electrophoretic mobility shift assay (EMSA) with reconstituted nucleosomes was carried out, and the replicates are presented below. Reproducible results were obtained, suggesting that BD was necessary and sufficient for its interaction with reconstituted nucleosomes. Further, in Supplementary Fig. 6, the native-ChIP assay was replaced with native-ChIP quantitative-PCR and shown as Fig. 6h.

R3-7: Low cell numbers (<100 per biological replicate) and a lack of justification for why particular cells were selected for analysis further raise potential issues with data analysis that must be addressed.

Au: A) The outer kinetochore in *C. neoformans*, as described previously and in this study (Fig. 2l and m), localizes to the kinetochore transiently at G2-M. Moreover, outer kinetochore

signals of the KMN network and Dam1 complex reduce post anaphase onset (Fig. 2l). Kinetochores are highly dynamic at anaphase, where cells spend only ~3 min in *C. neoformans* (Supplementary Fig. 5f). Therefore, we attempted to capture the variation in signal intensities of kinetochores, if any, at stages where their signal intensities peak, which is at metaphase (budding index ~0.55-0.8) for the KMN network components and Dam1 complex proteins. To maintain uniformity bridging was also measured at metaphase. It is noteworthy that centromeres in *C. neoformans* are present in an unclustered state in most of interphase³ (Supplementary Fig. 1k). While inner kinetochores, CENP-A^{Cse4} and CENP-C^{Mif2}, are constitutively localized to the kinetochores, the measurement of kinetochores signal intensities from these unclustered centromeres is technically challenging to perform for a large number (*n*) of cells. Clustered kinetochores in the daughter bud (budding index of ~ 0.55-0.8) gave us a reliable stage to score for the kinetochores intensities. Hence, we restricted to measuring kinetochores signals at this stage whenever possible. This explanation has been incorporated into the methods section from page 23 lines 3-9.

B) On the other hand, we agree with the reviewer's concerns regarding the optimum number (*n*) of cells required to examine for drawing a meaningful conclusion. Therefore, as suggested by the reviewer, we revised our kinetochores signal intensity measurements to include at least 100 cells/individual clustered kinetochores for each of the three independent biological replicates in Fig. 2c-f, 4d, 6c and Supplementary Fig. 3a-j, 7e-l in the revised manuscript. Each dot in the above-mentioned figures represents a single measurement and the red dots represent the mean of one biological replicate.

R3-8: Regarding interdependency analyses, low cell numbers for strains using the GAL inducible system, coupled with arbitrary imaging times and the lack of evidence of glucose-mediated repression following galactose induction in *Cryptococcus grubii* invalidate findings using this reporter system. The authors must demonstrate that protein expression is in fact repressed, for example through biological consequence of repression (mis-segregation analysis, altered growth, sensitivity phenotypes). They should include imaging times for the indicated strains in the figure legends, as well as rationale for those times in the main text. Note that Wickes et al., who developed this reporter, did so using *C. neoformans* var. *neoformans* and did not demonstrate that it is repressible following induction, only that it is not induced in the absence of galactose. Others using this system in *Cn. grubii* have observed leaky expression in glucose and have observed differences in induction (Baker and Lodge 2012).

Au: We thank the reviewer for raising this point. We have previously described the stringency of the *GAL7p* promoter (*GAL7p*) in *C. neoformans* var. *grubii* using the mitotic kinase AuroraB^{Pl1}⁴, and cited this reference with the explanation of the experimental system in the revised manuscript (page 7 lines 10-13 and page 22 lines 13-20).

A) Further, as suggested, we performed phenotype analyses, including spotting and viability assays, to show the effective shut-off of the *GAL7p* for the conditional kinetochores proteins of CENP-A^{Cse4}, CENP-C^{Mif2}, Mis12^{Mtw1}, Nuf2, Knl1^{Spc105}, Dad1 and Dad2 in *C. neoformans* (new Supplementary Fig. 2f and g). A lack of growth and increased loss in viability in the non-permissive conditions for these mutants of essential genes clearly indicated an essential function of the tested kinetochores proteins in *C. neoformans*. Additionally, using CENP-A^{Cse4} and Dad1, through immunoblot assays, we show that the protein levels are in fact reduced upon promoter shut-off (new Supplementary Fig. 2d and e). These results in addition to previous validation of the *GAL7p*-mediated repression of genes suggest the effective shut-off

of target genes under the *GAL7p* in non-permissive conditions, which result in an observed phenotype for deficiency of target proteins. Our data also suggest that the time taken to observe a phenotype is largely dependent on the stability and effective protein levels in the cell required to call out its function.

B) As we mentioned in response to comment R3-7, we have increased cell numbers for the kinetochore intensity measurements for interdependency analysis in revised Fig. 2c-e and Supplementary Fig. 3a-j, to 100 each for three independent experiments. Each dot in these figures represents a single measurement and the red dots represent the mean of one biological replicate.

C) The imaging time used for each mutant has been mentioned in the figure legends of Fig. 2 and Supplementary Fig. 3 in the revised manuscript. The imaging times were determined as the least time taken for strains to exhibit no visible puncta kinetochore signals upon culturing in the non-permissive conditions. This phenotype often corresponded to the time required to lose ~90% cell viability for the corresponding strain (new Supplementary Fig. 2g). This explanation for using the mentioned times has been described in the methods section Page 22 lines 16-20 in the revised manuscript.

Reviewer 4

Sridhar and colleagues show that several fungal lineages lack known kinetochore linker complexes and use proteomics to identify possible novel linker proteins in the basidiomycete *C. neoformans*. They find several candidates that were present in all pulldown experiments of three known kinetochore components, and zoom in on one based on its co-localization with a kinetochore marker. This candidate they name bridgin and the remainder of the manuscript details its functional and biochemical characterization. In short:

- its localization to kinetochores depends on the ndc80 complex
- its deletion from the *Cryptococcus* genome causes chromosome segregation errors
- both of its definable features (FHA domain, basic domain) are required for function
- Its basic domain binds aspecifically to DNA
- function of the basic domain depends not on its sequence but on its charge.

This is an interesting study that combines observation from comparative genomics (loss of CCAN complexes) with experimental investigations to explain those losses (identification and characterization of a new kinetochore protein). The data convincingly show bridgin is a kinetochore protein and that it is important for accurate chromosome segregation. The manuscript could do however with more insight into bridgin's molecular function.

R4-1: My main comment is that bridgin is presented as a bridge between chromatin and the outer kinetochore, but its function is actually quite poorly examined. For example, loss of known linker complexes in budding yeast or animal cells prevent the outer kinetochore from assembling properly. In bridgin's case, that is not examined.

Au: We appreciate the reviewer for raising this comment. A similar point was raised by R2. As we responded to the comment R2-1, we altered our description of bridgin by not referring to a "kinetochore linker" protein.

Further, as mentioned in response to reviewer comments R2-6 and R3-7, we performed additional experiments and measured, the effect of bridgin on several key kinetochore subunits of CENP-A^{Cse4}, CENP-C^{Mif2}, Mis12^{Mtw1}, Nuf2, and Knl1^{Spc105} in metaphase and

anaphase ($n = 100$ for each of three independent experiments) (new Supplementary Fig. 7e-i). We find that certain outer, but not inner, kinetochore proteins are reduced by a small but significant extent upon *bgi1Δ*. Yet, we suspect it is premature to conclude and would require additional biochemical experiments which we believe is beyond the scope of this first description and characterization of a novel family of bridgin proteins.

R4-2: In fact, bridgin depends on the Mis12- and Knl1 complexes (not sure why they conclude that it depends on the Ndc80 complex (fig 2g): that is not in fact tested).

Au: We thank the reviewer for pointing it out. Although we had mentioned in the main text stating that bridgin is dependent on Mis12C and the Knl1C protein Sos7 (Previous version, Page 7, line 4 “On the other hand...”) we believe that the schematic presented in Figure 2g may be confusing and was redrawn in the revised manuscript. We observe that the Mis12C and the Ndc80C are dependent on each other in *C. neoformans* (New Supplementary Fig. 3a and b) and the Mis12C conditional mutant reduced bridgin levels to below detection (Fig. 2d). Due to the interdependency of the Mis12C and Ndc80C on each other, it would be technically difficult to determine the contribution of the individual complexes on bridgin localization by our interdependency assay. Yet, it is also likely, given the interdependency of Ndc80C and Mis12C on each other, that a conditional mutant of the Ndc80C would affect bridgin levels, as represented in the schematic (Fig. 2g). In the revised manuscript, we present data to suggest that bridgin localization is in fact, affected by Mis12C and Knl1C mutants (new Fig 2d-f).

R4-3: If the KMN network does not depend on bridgin, it can hardly be a linker complex. In fact, it would mean that *Cryptococcus*’ most essential linker ‘activity’ is still mysterious.

Au: Although we show, as mentioned in response to comment 4-1, that certain KMN network components are significantly reduced in a bridgin null mutant, we do not conclude that bridgin is a linker protein with only the current data. Agreeing with this comment from this reviewer and similar comments from R2, we modified the text by not terming bridgin as a linker protein in the revised manuscript.

Further, we agree with the reviewer that the precise mode of bridgin action has not been completely addressed in this study. Instead, we identify the previously undescribed kinetochore protein bridgin and by functionally dissecting its domains, hypothesize a function for it at the *C. neoformans* kinetochore (new Fig. 7a). We show that the ability of bridgin to interact with centromeric chromatin through its basic domain following its recruitment to the outer kinetochore is critical for its function to mediate accurate chromosome segregation. We believe future studies to examine the precise contribution of the bridgin pathway at the kinetochore, not only in *C. neoformans* but also in other systems where it has been identified are required.

R4-4: Then there is the observation that bridgin binds DNA and chromatin aspecifically. It is not easy to imagine how that is a useful characteristic of a linker protein. Does it bind centromeric sequences or chromatin with higher affinity than non-centromeric ones? Does it need DNA binding? The fact that deleting BD compromises function and can be replaced with the BD of Ki67 is in my view not sufficient evidence that it is the DNA-binding capacity of BD that is needed for bridgin function.

Au: We thank the reviewer for raising these relevant points.

A) The points raised here are similar to the R2-5 from R#2. As we responded, with additional data, we show that the basic domain of bridgin can indeed interact with chromatin *in vivo*. Based on the series of evidence gathered from over-expression assays (Fig. 5g), co-IP (new Fig. 6g and Supplementary Fig. 6b), domain swap (Fig. 6d and e) and native ChIP experiments (new Fig. 6h), we convincingly demonstrate that the BD of bridgin interacts with centromeric chromatin and that it is required for bridgin's function.

B) Our *in vivo* results presented in Figures 5g, 6a-e, and 6g and h of the revised manuscript show that the interaction of BD with chromatin is non-specific and not dependent on its amino acid sequence. Yet, Bgi1^{FL} specifically interacts with centromeric chromatin *in vivo*. Thus, we believe that the recruitment of Bgi1^{FL} onto centromeres occurs through the outer kinetochore proteins ensures its specific interaction with centromeric chromatin through BD. We further displayed that over-expression of Bgi1^{FL} or Bgi1^{BD} can mislocalize bridgin to bulk chromatin, suggesting that the overall cellular levels of bridgin are regulated to prevent mislocalization. However, we cannot completely rule out the possibility of the existence of additional factors restricting BD-centromeric chromatin interactions. We described these points in the revised text.

R4-5: In addition: most elements of the model in 6g are not shown: does the FD bind outer kinetochore proteins? Does the BD bind centromeric DNA?

Au: In the revised manuscript, we present additional data that supports our model:

A) Additional interdependency assays with Knl1, as described in response to comment R1-1B, is now presented. Our data suggest that FD and USD can localize to the kinetochore independently. Further, we have modified the model to place less emphasis on a specific outer kinetochore-bridgin interaction due to limited biochemical data to support this interaction.

B) We, as described in response to comment R4-4 and R2-5, show that BD in Bgi1^{FL} can bind to centromere DNA based on native ChIP combined with qPCR.

R4-6: What can be a molecular explanation for why BD does not bind DNA unless FD binds the kinetochore?

Au: This is an interesting question and we speculate three possibilities:

A) Phospho-regulation of bridgin BD: The basic domain of Ki67, a possible bridgin homolog, is hyperphosphorylated at interphase, thereby altering its overall charge at the end of mitosis and preventing its interaction with chromatin^{7,8}.

B) Regulation of bridgin protein levels through the cell cycle: We tested and ruled out this possibility as we found relatively consistent levels of bridgin through interphase and M phase (new Supplementary Fig. 2b).

C) Structural changes induced by the outer kinetochore binding of bridgin facilitating DNA binding by BD.

R4-7: Bridgin is proposed to promote stable kinetochore-microtubule interactions, but that is inferred mostly from the observations that bridgin mutants delay in mitosis in a mad2-

dependent manner and that they show missegregation. These can be explained by many other possible defects.

Au: We agree with this comment and cannot completely rule out other possible explanations. However, we showed substantial SAC-mediated delay, segregation defects, and sensitivity to spindle insults in bridgin mutants. In light of the absence of molecular tools to address the precise nature of kinetochore-microtubule interactions in *C. neoformans*, to address the concerns of the reviewer, we toned down the emphasis on the effect of bridgin on kinetochore-microtubules and rephrased the functions of bridgin by stating that it is required for mitotic progression and accurate chromosome segregation. We responded similarly to the comment R1-4 and R2-7.

R4-8: In short, I feel the finding is potentially interesting but quite preliminary at this stage.

Au: We incorporated suggestions from all reviewers as much as possible, and we substantially revised the manuscript with additional experimental data. We now hope that the revised manuscript adequately addressed all the concerns and is now suitable for publication in *Nature Communications*.

References:

1. Jakopiec, V., Topolski, B. & Fleig, U. Sos7, an Essential Component of the Conserved Schizosaccharomyces pombe Ndc80-MIND-Spc7 Complex, Identifies a New Family of Fungal Kinetochore Proteins. *Mol. Cell. Biol.* **32**, 3308–3320 (2012).
2. Hooff, J. J., Tromer, E., Wijk, L. M., Snel, B. & Kops, G. J. Evolutionary dynamics of the kinetochore network in eukaryotes as revealed by comparative genomics. *EMBO Rep.* **18**, 1559–1571 (2017).
3. Kozubowski, L. *et al.* Ordered kinetochore assembly in the human-pathogenic basidiomycetous yeast *Cryptococcus neoformans*. *MBio* **4**, (2013).
4. Varshney, N. *et al.* Spatio-temporal regulation of nuclear division by Aurora B kinase Ipl1 in *Cryptococcus neoformans*. *PLoS Genet.* **15**, e1007959 (2019).
5. Vleugel, M., Hoogendoorn, E., Snel, B. & Kops, G. J. P. L. Perspective Evolution and Function of the Mitotic Checkpoint. (2012). doi:10.1016/j.devcel.2012.06.013
6. Kops, G. J. P. L., Snel, B. & Tromer, E. C. Evolutionary Dynamics of the Spindle Assembly Checkpoint in Eukaryotes. *Curr. Biol.* **30**, R589–R602 (2020).
7. Endl, E. & Gerdes, J. Posttranslational modifications of the KI-67 protein coincide with two major checkpoints during mitosis. *J. Cell. Physiol.* **182**, 371–380 (2000).
8. MacCallum, D. E. & Hall, P. A. Biochemical Characterization of pKi67 with the Identification of a Mitotic-Specific Form Associated with Hyperphosphorylation and Altered DNA Binding. *Exp. Cell Res.* **252**, 186–198 (1999).

REVIEWER COMMENTS

Reviewer #1 (Remarks to the Author):

The reviewer have read the revised manuscript and the rebuttal letter and found that the authors have addressed all my concerns. The manuscript has been improved significantly after comprehensive revision and the reviewer think it is appropriate for publication in Nature Communications.

Reviewer #2 (Remarks to the Author):

My major issue with the original manuscript was to call bridgin a linker kinetochore protein. In the revised manuscript, the authors modified the text to not to associate the term kinetochore linker with bridgin. However, although they changed the title to "Bridgin connects the outer kinetochore to centromeric chromatin", it could still give an impression that bridgin is important for recruiting outer kinetochore proteins (they observed very subtle reduction in KMN signals in *bgi1Δ* cells). I therefore suggest that the title is changed to something like "Bridgin associates with centromeric chromatin and the outer kinetochore". Similarly, they may also change the abstract "its ability to connect the outer kinetochore with centromeric chromatin (page 2, line 9)" appropriately. Other than this point, I do not have any further issue and I remain supportive of the publication of this manuscript in Nature Communications.

Reviewed by Bungo Akiyoshi

Reviewer #3 (Remarks to the Author):

The authors have addressed my concerns.

Reviewer #4 (Remarks to the Author):

The manuscript is much improved, and I in principle support its publication if the following issues are addressed:

1. The minor contribution of *Sos7/spc105* to *Bgi1* kinetochore localization is now represented as one of two arms for *Bgi1* recruitment (Fig. 2g). One is absolutely required (*Mis12C*), the other only so-so (*Spc105/Sos7*). It seems likely to this reviewer that deletion of *Sos7* or *Spc105-OFF* compromises the *Mis12* complex, thus indirectly affecting *Bgi1* localization. This would explain the current data more satisfactorily than the 2-arm hypothesis. The human kinetochore field has gone through similar situations that caused quite a bit of confusion. Partial effects of depletion of kinetochore protein X on localization of protein Y, interpreted as a non-essential contribution of X to Y recruitment. Later it was often clarified that depletion of protein X destabilized certain kinetochore complexes, indirectly affecting protein Y recruitment (e.g. depletion of the human *Sos7* homolog *zwint* indirectly affects

RZZ recruitment by destabilizing the KNL1-Mis12 complexes, while Zwint was originally thought to directly recruit RZZ). I would recommend the authors include one additional experiment to clarify the recruitment hierarchy: assess Mis12C in the Sos7-delta and Spc105-OFF strains. If reduced, the cartoon in 2g can be simplified, saving the field some potential confusion. If not reduced, the scheme can stay as is.

2. I recommend changing the name bridgin. Multiple reviewers raised concerns about the interpretation of Bgi1 as a linker protein, and now these interpretations have been toned down. I think the name will lead to confusion with readers and possibly to mis-interpretation, as it eludes to a linker/bridge function, which the authors themselves say is to be examined still. The authors already have an alternative name (Bkt1).

Geert Kops

Point-by-point response

Sridhar *et al.*

Nature Communications (MS# NCOMMS-19-32938A)

Reviewer #1 (Remarks to the Author):

R1-1: The reviewer has read the revised manuscript and the rebuttal letter and found that the authors have addressed all my concerns. The manuscript has been improved significantly after comprehensive revision and the reviewer think it is appropriate for publication in Nature Communications.

Au: We thank the reviewer for being supportive.

Reviewer #2 (Remarks to the Author):

Au: My major issue with the original manuscript was to call bridgin a linker kinetochore protein. In the revised manuscript, the authors modified the text to not to associate the term kinetochore linker with bridgin. However, although they changed the title to “Bridgin connects the outer kinetochore to centromeric chromatin”, it could still give an impression that bridgin is important for recruiting outer kinetochore proteins (they observed very subtle reduction in KMN signals in *bgi1Δ* cells). I therefore suggest that the title is changed to something like “Bridgin associates with centromeric chromatin and the outer kinetochore”. Similarly, they may also change the abstract “its ability to connect the outer kinetochore with centromeric chromatin (page 2, line 9)” appropriately. Other than this point, I do not have any further issue and I remain supportive of the publication of this manuscript in Nature Communications.

Reviewed by Bungo Akiyoshi

R2-1: We thank Dr. Akiyoshi for his positive remarks. While we appreciate the minor concerns raised by the reviewer and possible modifications sought in the title, we feel that the current title accurately highlights the findings in the manuscript. We established beyond doubt that bridgin, being recruited to the outer kinetochore KMN network, has the ability to subsequently interact with centromeric chromatin justifying the correct title. Since we find a subtle reduction in levels of outer kinetochore proteins upon bridgin deletion, we do not claim that bridgin is responsible for their recruitment. Taken together our findings point to bridgin interacting with centromeric chromatin following its recruitment to the outer kinetochore thus exhibiting its ability to connect the outer kinetochore to centromeric chromatin, the two spatially linked but distinct entities.

Reviewer #3 (Remarks to the Author):

R3-1: The authors have addressed my concerns.

Au: We thank the reviewer for the positive response.

Reviewer #4 (Remarks to the Author):

R4: The manuscript is much improved, and I in principle support its publication if the following issues are addressed:

1. The minor contribution of *Sos7/spc105* to *Bgi1* kinetochore localization is now represented as one of two arms for *Bgi1* recruitment (Fig. 2g). One is absolutely required (*Mis12C*), the other only so-so (*Spc105/Sos7*). It seems likely to this reviewer that deletion of *Sos7* or *Spc105-OFF* compromises the

Mis12 complex, thus indirectly affecting Bgi1 localization. This would explain the current data more satisfactorily than the 2-arm hypothesis. The human kinetochore field has gone through similar situations that caused quite a bit of confusion. Partial effects of depletion of kinetochore protein X on localization of protein Y, interpreted as a non-essential contribution of X to Y recruitment. Later it was often clarified that depletion of protein X destabilized certain kinetochore complexes, indirectly affecting protein Y recruitment (e.g. depletion of the human Sos7 homolog *zwint* indirectly affects RZZ recruitment by destabilizing the KNL1-Mis12 complexes, while *Zwint* was originally thought to directly recruit RZZ). I would recommend the authors include one additional experiment to clarify the recruitment hierarchy: assess Mis12C in the *Sos7-delta* and *Spc105-OFF* strains. If reduced, the cartoon in 2g can be simplified, saving the field some potential confusion. If not reduced, the scheme can stay as is.

Au: We thank Dr. Kops for the overall positive response towards our revised manuscript and for raising this point. The experiment suggested by the reviewer concerning the influence of Knl1C on Mis12C has already been described in our manuscript (Supplementary Fig. 3d), wherein we have performed and quantitated the interdependency between *Kn1^{Spc105}* (Knl1C) and *Mis12^{Mtw1}* (Mis12C). Upon *Kn1^{Spc105}-OFF* we show that levels of Mis12 are not reduced. Thus, suggesting that the Knl1C does not compromise the levels of Mis12C at the kinetochore in the conditional mutant.

Consequently, the observation of bridgin dependence on Knl1C (Fig. 2e and f) indeed describes one of two arms of bridgin recruitment. It is possible that upon *Mis12C-OFF*, *Mis12C* influencing *Kn1C*, a combined loss of *Mis12C* and *Kn1C* kinetochore localization may result in a synergistic affect resulting in complete loss of bridgin kinetochore localization. Taken together we show that bridgin recruitment to the kinetochore is dependent on the *Mis12C* and *Kn1C* complexes.

Given the prevailing circumstances it would be challenging to test the influence of *Sos7* (*Kn1C* component) on *Mis12C*. However, we believe that *Mis12^{Mtw1}* in a *Kn1^{Spc105}* (*Kn1C* component) conditional mutant (Supplementary Fig. 3d) well represents the dependency of the *Mis12C* on the *Kn1C*.

2. I recommend changing the name bridgin. Multiple reviewers raised concerns about the interpretation of *Bgi1* as a linker protein, and now these interpretations have been toned down. I think the name will lead to confusion with readers and possibly to mis-interpretation, as it eludes to a linker/bridge function, which the authors themselves say is to be examined still. The authors already have an alternative name (*Bkt1*).

Geert Kops

Au: In this manuscript we describe the identification of bridgin as a kinetochore protein that is recruited to the outer kinetochore, which requires its FHA and unstructured domains (Fig. 2d-f and Fig. 4c). Following bridgin's kinetochore recruitment we show that through its basic domain bridgin has the ability to interact with centromeric chromatin (Fig. 6g and h). We also observe that bridgin's kinetochore localization is not influenced by its chromatin binding (Fig. 4c). Further, we describe the subtle but significant reduction in the levels of KMN network proteins in the bridgin-null mutant. Yet we are cautious, also taking into the reviewer's comments, do not conclude that bridgin recruits outer kinetochore protein and performs a linker function without further evidence (Fig. 7a and Supplementary Fig. 7g-i).

Taken together with multiple lines of evidence, we established that an unknown protein bridgin is localized to the outer kinetochore while being able to interact with centromeric chromatin. Thus, the role played by bridgin to connect/bridge two spatially linked but distinct entities is an uncommon feature amongst kinetochore proteins. Through our presented *in vitro* and *in vivo* experimental evidence, we are confident of the connecting/bridge ability of bridgin. We, in the manuscript suggest that further investigation is required to understand the contribution of the described connecting/bridging ability of bridgin towards kinetochore function and not on the connection/bridge ability itself. Thus, we believe that the naming of "bridgin" accurately reflects the connecting/bridging property of the FHA domain containing long unstructured region, as described through our experimental evidence and renaming it to non-functional name would not be correct.

Our initial finding that loss of bridgin leads to a subtle but significant reduction in the levels of outer kinetochore proteins (Supplementary Fig. 7g-h) may in subsequent studies describe the importance of the connecting/bridge function of bridgin.

REVIEWERS' COMMENTS

Reviewer #4 (Remarks to the Author):

The authors satisfactorily responded to my request for the assessment of mis12C in the absence of knl1, which indeed argues for two arms of bridgin recruitment. As for the name: i guess we can agree to disagree. In my view, a bridging function is vital to connect A to B, which is not shown for bridgin. Nevertheless, I'm fine with leaving it as is, and will follow with great interest the future research by this group on uncovering mechanisms of bridgin function.

Point-by-point response

Sridhar *et al.*

Nature Communications (MS# NCOMMS-19-32938B)

Reviewer #4 (Remarks to the Author):

R4-1: The authors satisfactorily responded to my request for the assessment of mis12C in the absence of knl1, which indeed argues for two arms of bridgin recruitment. As for the name: i guess we can agree to disagree. In my view, a bridging function is vital to connect A to B, which is not shown for bridgin. Nevertheless, I'm fine with leaving it as is, and will follow with great interest the future research by this group on uncovering mechanisms of bridgin function.

Au: We thank Prof. Kops for his appreciative and encouraging remarks. We indeed are excited and looking forward to build on our bridgin work presented in this study.